# Gαq-PKD/PKCμ signal regulating the nuclear export of HDAC5 to induce the IκB expression and limit the NF-κB-mediated inflammatory response essential for early pregnancy

Yufei Jiang[1]*[†], Yan He[2][†], Songting Liu[3], Gaizhen Li[3], Dunjin Chen[4], Wenbo Deng[3], Ping Li[1], Ying Zhang[3], Jinxiang Wu[3], Jianing Li[3], Longmei Wang[1], Jiajing Lin[5], Haibin Wang[3]*, Shuangbo Kong[3]*, Guixiu Shi[2]*

[1]Xiamen Key Laboratory of Reproduction and Genetics, Department of Reproductive Medicine, Women and Children's Hospital, School of Medicine, Xiamen University, Xiamen, China; [2]Xiamen Key Library of Rheumatology and Clinical Immunology, Department of Rheumatology and Clinical Immunology, The First Affiliated Hospital of Xiamen University, School of Medicine, Xiamen University, Xiamen, China; [3]Fujian Provincial Key Laboratory of Reproductive Health Research, Department of Obstetrics and Gynecology, The First Affiliated Hospital of Xiamen University, School of Medicine, Xiamen University, Xiamen, China; [4]Department of Pathology, Women and Children's Hospital, School of Medicine, Xiamen University, Xiamen, China; [5]Department of Obstetrics and Gynecology, Key Laboratory for Major Obstetric Diseases of Guangdong Province, The Third Affiliated Hospital of Guangzhou Medical University, Guangzhou, China

*For correspondence:
ji.angyufei@163.com (YJ);
haibin.wang@vip.163.com (HW);
shuangbo_kong@163.com (SK);
gshi@xmu.edu.cn (GS)

[†]These authors contributed equally to this work

Competing interest: The authors declare that no competing interests exist.

**Abstract** Decidualization, denoting the transformation of endometrial stromal cells into specialized decidual cells, is a prerequisite for normal embryo implantation and a successful pregnancy in human. Here, we demonstrated that knockout of Gαq lead to an aberrantly enhanced inflammatory state during decidualization. Furthermore, we showed that deficiency of Gαq resulted in over-activation of nuclear factor (NF)-κB signaling, due to the decreased expression of *NFκBIA*, which encode the IκB protein and is the negative regulator for NF-κB. Mechanistically, Gαq deficiency decreased the Protein kinase D (PKD, also called PKCμ) phosphorylation levels, leading to attenuated HDAC5 phosphorylation and thus its nuclear export. Aberrantly high level of nuclear HDAC5 retarded histone acetylation to inhibit the induced *NFκBIA* transcription during decidualization. Consistently, pharmacological activation of the PKD/PKCμ or inhibition of the HDAC5 restored the inflammatory state and proper decidual response. Finally, we disclosed that over-active inflammatory state in Gαq-deficient decidua deferred the blastocyst hatching and adhesion in vitro, and the decidual expression of Gαq was significantly lower in women with recurrent pregnancy loss compared with normal pregnancy. In brief, we showed here that Gαq as a key regulator of the inflammatory cytokine's expression and decidual homeostasis in response to differentiation cues, which is required for successful implantation and early pregnancy.

## Editor's evaluation

This study presents a valuable finding on the role and function of the Gαq axis on the inflammatory response during decidualization essential for early pregnancy. The evidence supporting the claims of the authors is solid and clarified. The work will be of interest to reproductive biologists and clinicians.

## Introduction

Successful pregnancy requires precisely coordinated immune responses between the mother and fetus (*Yockey and Iwasaki, 2018*). Disorders in these communications are tightly associated with adverse pregnancy outcomes including recurrent pregnancy loss (RPL) (*Wu et al., 2018*; *Yockey and Iwasaki, 2018*). Increasing evidence has demonstrated compromised inflammatory cytokine expression in the endometrium from RPL patients (*Wu et al., 2018*; *Wang et al., 2021*). Cytokine signaling pathways participate the differentiation of endometrium to promote healthy pregnancy, for their ability to drastically alter cellular function, cell–cell communication, and gene expression (*Vinketova et al., 2016*). However, when dysregulated or inappropriately expressed during the inflammation, cytokines have the potential to act as teratogens and disrupt fetal and placental developmental, leading to birth defects and pregnancy complications (*Yockey and Iwasaki, 2018*). Investigating the molecular mechanisms to regulate the inflammatory cytokine's expression, which impact pregnancy and fetal development, could provide important insights into congenital disorders and possible therapeutics to prevent pregnancy complications.

Cyclic decidualization, formed during the secretory phase of menstrual cycle from uterine mucosa termed endometrium, is crucial for embryo implantation and human reproductive success (*Gellersen and Brosens, 2014*). Decidual stromal cells (DSCs) differentiate from endometrial stromal cells (ESCs) under the influence of ovarian hormones and cyclic adenosine monophosphate (cAMP). DSCs acquire specific functions related to considerable transcriptional and cellular remodeling, enabling implantation and placental development as well as establishment of maternal immune tolerance (*Evans et al., 2016*; *Vinketova et al., 2016*). At the molecular level, decidual transformation of ESCs involves genome-wide remodeling of the chromatin landscape (*Vrljicak et al., 2018*), wholesale reprogramming of several signaling pathways (*Cloke et al., 2008*; *Leitao et al., 2010*), including progesterone and cAMP/protein kinase A (PKA) signaling pathways. These signals induce a downstream cascade (*Pavličev et al., 2017*, *Wu et al., 2018*) and transcription factors, such as progesterone receptor (PGR), forkhead box protein O1 (FOXO1), homeobox (HOX), CCAAT/enhancer-binding protein $\beta$ (C/EBP$\beta$), and signal transducers and activators of transcription (STATs) paralogs (*Gellersen and Brosens, 2014*; *Vinketova et al., 2016*), that together set in motion for decidualization regulatory program. Progesterone and oestrogen, detached from decidual environment, are not strong stimulators of decidualization. This implies that other signal molecule, such as prokineticin-1 (*Cook et al., 2010*), relaxin (*Tseng et al., 1992*), and prostaglandin E2 (PGE2) (*Dimitriadis et al., 2005*) from the endometrial niche synergistically augment decidual transformation of ESCs. The receptors of prokineticin-1, relaxin, and PGE2 are G-protein-coupled receptors (GPCRs) characterized by the presence of seven transmembrane spanning domains (*Cook et al., 2010*) that are primarily considered to couple Gαs to activate adenylate cyclase that subsequently forms the second messenger cAMP. cAMP signaling participates the decidualization by the activation of PKA to influence the expression of major decidual factors, including prolactin (PRL) (*Telgmann et al., 1997*) and insulin-like growth factor-binding protein-1 (IGFBP-1) (*Tamura et al., 2012*), two specific markers of ESC decidualization.

Gαq, encoded by the GNAQ gene, belongs to the Gq/11 subfamily of heterotrimeric G proteins, and it is ubiquitously expressed in mammalian cells (*Wilkie et al., 1991*). The Gαq links GPCRs to activate Phospholipase C (PLC) that hydrolyzes membrane phospholipids phosphatidylinositol bisphosphate, PIP2 to release inositol trisphosphate (IP3) and diacylglycerol (DAG) – two important second messengers causing $Ca^{2+}$ mobilization and protein kinase C (PKC) activation (*Ho et al., 2009*). The activated molecules each participate in multiple signaling networks involved in inflammatory responses, and link Gαq family members to a host of different cellular events. As a critical downstream effector for GPCR, whether and how the Gαq-mediated signaling controls decidualization still need investigation. In the present study, we showed that Gαq played an important role in balancing the inflammatory cytokine expression and decidual progression in response to deciduogenic cues.

## Results

### Gαq is expressed in human endometrium, decidua of early pregnancy, and DSCs in vitro

The expression of Gαq protein in human endometrial tissues during menstrual cycle and in decidual tissues of early pregnancy was detected by immunohistochemistry. The positive expression of Gαq showed in the endometrium of mid-proliferative, mid-secretary phase, and decidual tissue of early pregnancy (*Figure 1—figure supplement 1A*). Expression of Gαq was higher in stromal cell at mid-secretary phase during menstrual cycle, which is the timing for implantation (implantation windows), and in deciduas of early pregnancy (*Figure 1—figure supplement 1A*). These results imply that Gαq may play an important role during stromal cell decidualization which occurred in implantation windows and early pregnancy. To identify the pathophysiological relevance of Gαq to human endometrial decidualization, Gαq expression was further examined in the proliferative and differentiated human endometrial stromal cell (HESC) in vitro. As shown in *Figure 1—figure supplement 1B, C*, the expression of both the Gαq transcript and protein increased significantly in the HESC as the differentiation progressed over time in vitro. Recently, GPCRs, G proteins, and their downstream effectors have been uncovered on the nuclear membrane or in the nucleus (*Campden et al., 2015*). So, with regard to the protein localization, immunofluorescence analyses and subcellular fraction experiments were performed to confirm the localization of Gαq in decidual cells. As shown in *Figure 1—figure supplement 1D, E*, Gαq was expressed in both cytoplasm and nucleus of proliferative and differentiated cells. This observation was further confirmed in primary cultured ESCs (*Figure 1—figure supplement 1F*).

### Gαq deficiency causes aberrant progression of decidualization with high expressed decidual markers

To explore the role of Gαq during decidualization, we utilized CRISPR/Cas9 technology to knock out endogenous *GNAQ* in HESC, and generated GNAQ-KO cell lines (*Figure 1—figure supplement 2*). Sanger sequencing proved that the clonal cell line harbors a single base insertion near the Cas9 cutting site, causing a frameshift in the protein coding region (*Figure 1—figure supplement 2A*), and this resulted in loss of endogenous Gαq expression (*Figure 1—figure supplement 2B*). We then interrogated the significance of Gαq during stromal cell decidualization by detecting the expression of Ki67, a proliferation marker. It showed no significant change in control (CON) and GNAQ-KO HESC in the proliferative state, but remarkable decrease in GNAQ-KO cells on the first day of decidua induction (*Figure 1A*). According to previous studies, HESC usually stopped proliferating on the fourth day of decidual induction (*Jiang et al., 2020*). Meanwhile, compared with control decidual cells, GNAQ-KO significantly accelerated the expression of *PRL* and *IGFBP1* mRNA (*Figure 1B, C*). Consistently, the level of IGFBP1 protein was also increased significantly upon Gαq deficiency (*Figure 1D*). Decidualization promotes profound changes of signaling pathways and expression of transcription factors (*Pavličev et al., 2017*, *Wu et al., 2018*). To identify novel downstream targets that contribute to Gαq-mediated progression of decidualization, RNA sequencing (RNA-seq) was utilized to explored the differently expressed genes in CON and GNAQ-KO decidual cells (differentiated for 3 days). More than 3880 genes were altered in the absence of GNAQ (*Figure 1E*). Kyoto Encyclopedia of Genes and Genomes (KEGG) analysis revealed that most top-ranked biological processes included the Wnt signaling pathway (*Sharma et al., 2016*; *Pathak et al., 2021*) and JAK–STAT, NF-κB inflammation pathway (*Figure 1F, G*), which had been reported to regulate stromal apoptosis and differentiation during the decidualization, the latter consistent with the notions regarding the role of Gαq during immune regulation (*Wang et al., 2014*; *Wang et al., 2017*).

### Gαq modulates the NF-κB signaling pathway and inflammatory status during decidualization

Precisely regulated inflammatory response is associated with the progression of decidualization (*Rytkönen et al., 2019*). The NF-κB pathway enrichment in the upregulated genes promoted us to explore its status in the absence of GNAQ, since it was also known to be activated by Gαq (*Xie et al., 2000*). As shown in *Figure 2A–C*, *Figure 2—figure supplements 1A and 2A–C*, nuclear translocation and phosphorylation of the NF-κB subunits p65 and p50 were higher in GNAQ-KO decidual cells

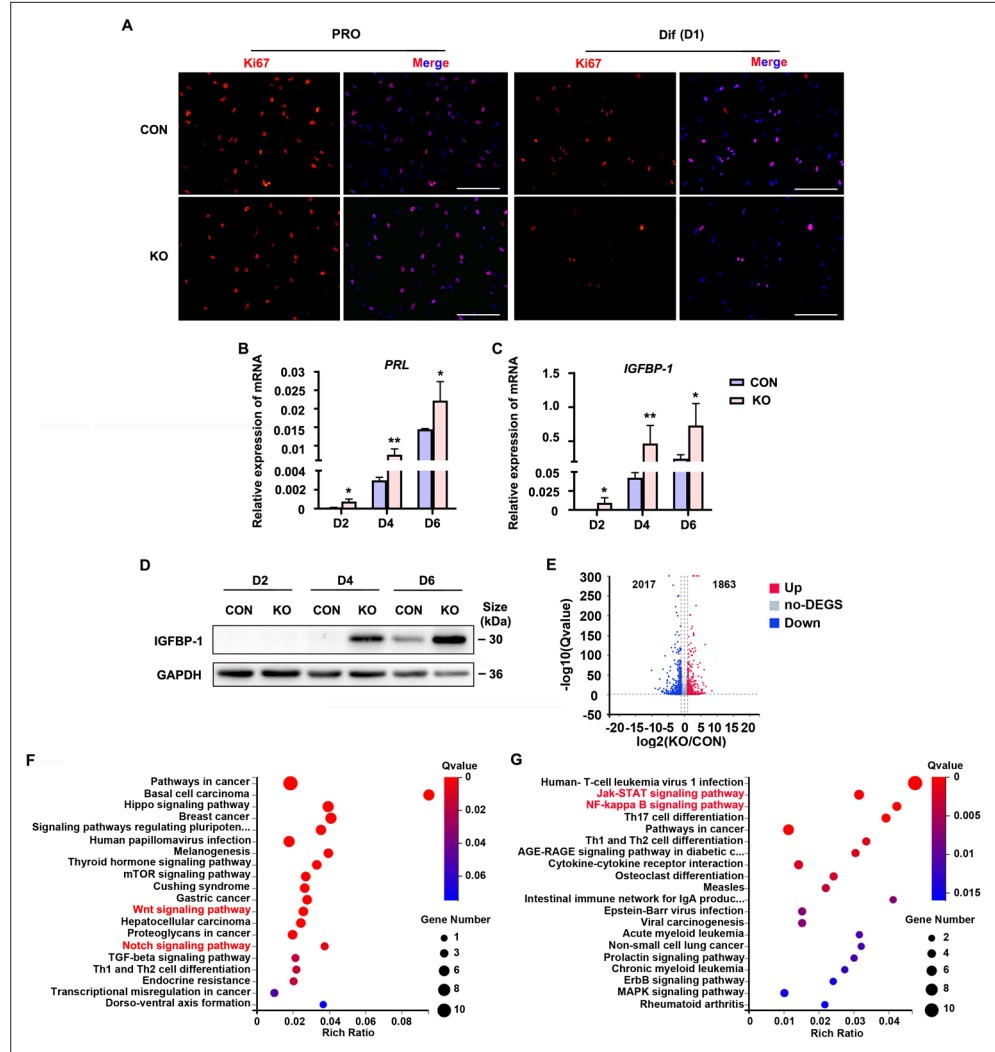

**Figure 1.** Human endometrial stromal cell (HESC) decidualization was advanced in the absence of Gαq. (**A**) Immunofluorescence staining of Ki67 in control (CON) and GNAQ knockout (KO) proliferative HESCs (D0) and decidual HESC treated with differential culture medium (Dulbecco's modified Eagle's medium [DMEM]/F12 with 2% charcoal-stripped fetal bovine serum [CS-FBS], dibutyryl cAMP [dbcAMP], and medroxyprogesterone 17-acetate (MPA) for 1 day. Scale bars, 100 µm. (**B, C**) Quantitative RT-PCR (qRT-PCR) analysis of prolactin (PRL) and insulin-like growth factor-binding protein-1 (IGFBP-1) mRNA expression levels in CON and KO HESC treated with differential culture medium for 2, 4, and 6 days (*n* = 5). (**D**) Western blot analysis of the IGFBP-1 protein levels in CON and KO decidual cells treated as in (**B**). GAPDH was used as the loading control. (**E**) Volcano plots showing significantly altered genes of KO/CON cells treated with differential culture medium for 3 days, UP, significant up regulated genes (*Q* value ≤0.05, red); Down, significant downregulated genes (*Q* value ≤0.05, blue); no-DEGS, no significant changed genes (gray). (**F, G**) Kyoto Encyclopedia of Genes and Genomes (KEGG) pathway analysis upregulated genes of in knockout cells during decidualization (*Q* value ≤0.05). Representative data are shown from three to five independent experiments. (**B**) and (**C**) were analyzed with unpaired Student's *t*-test. *p < 0.05 and **p < 0.01.

The online version of this article includes the following source data and figure supplement(s) for figure 1:

**Source data 1.** Original blot images of *Figure 1D*.

**Figure supplement 1.** Gαq is expressed in both human endometria in vivo and decidual stromal cells in vitro.

**Figure supplement 1—source data 1.** Original blot images of *Figure 1—figure supplement 1C and E*.

**Figure supplement 2.** Generation of GNAQ knockout human endometrial stromal cell (HESC) lines.

**Figure supplement 2—source data 1.** Original blot images of *Figure 1—figure supplement 2B*.

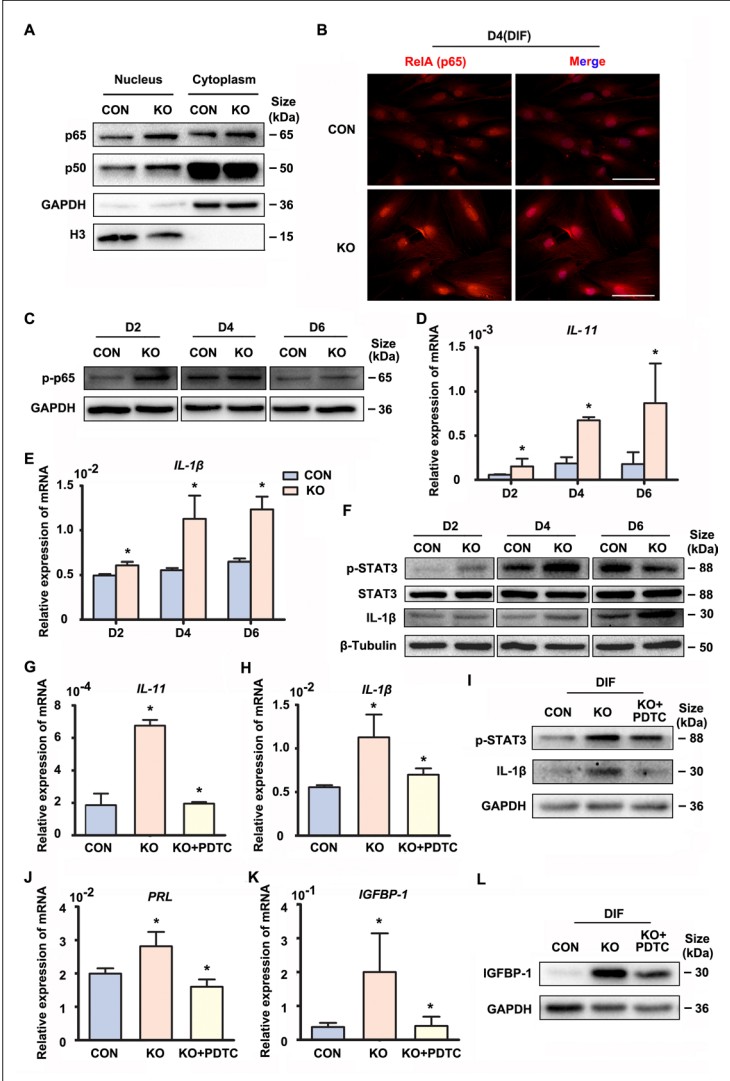

**Figure 2.** Gαq deficiency increased inflammatory cytokine expression in decidual stromal cells. (**A**) Western blot analysis of the p65 and p50 expression in cytoplasm and nucleus of CON and KO human endometrial stromal cell (HESC) treated with differential culture medium for 4 days, nuclear translocation of p65 and p50 were higher in GNAQ-KO decidual cells than in control. GAPDH and histone 3 (H3) were used as the loading controls. (**B**) Immunofluorescence staining of p65 in CON and KO stromal cells and treated with differential culture medium for 4 days, nuclear translocation p65 were higher in GNAQ-KO decidual cells than in control. Scale bars, 100 μm. (**C**) Western blot analysis of p-p65 expression in whole-cell lysates of CON and KO cells treated with differential culture medium for 2, 4, and 6 days, phosphorylation of the nuclear factor (NF)-$\kappa$B subunit p65 were higher in GNAQ-KO decidual cells than in control. GAPDH was used as the loading controls. (**D, E**) qRT-PCR analysis of *IL-11* and *IL-1β* mRNA expression levels in CON and KO HESC treated as in (**C**) (n = 5). (**F**) Western blot analysis of p-STAT3, STAT3, and IL-1β expression in whole-cell lysates of CON and KO HESC treated as in (**C**). β-Tubulin was used as the loading control. (**G, H, J, K**) qRT-PCR analysis of *IL-11*, *IL-1β*, *PRL*, and *IGFBP-1* mRNA after treatment with NF-$\kappa$B inhibitor, pyrrolidinedithiocarbamate ammonium (PDTC, 10 μM), in CON and KO HESC treated with differential culture medium for 4 days (n = 3). (**I, L**) Western blot analysis of p-STAT3, IL-1β, and IGFBP-1 expression in whole-cell lysates of differentiated CON and KO HESC treated as in (**G**). GAPDH was used as the loading control. Representative data are shown from three to five independent experiments. (**D and E**) were analyzed with unpaired Student's *t*-test. *p < 0.05 . (**G, H, J, and K**) were calculated with one-way analysis of variance (ANOVA) with Bonferroni's multiple comparison tests. *p < 0.05.

The online version of this article includes the following source data and figure supplement(s) for figure 2:

**Source data 1.** Original blot images of *Figure 2*.

*Figure 2 continued on next page*

*Figure 2 continued*

**Figure supplement 1.** Immunofluorescence staining of p50 and p65 in CON, KO or PDTC treated differentiated stromal cells.

**Figure supplement 2.** Relative of western blot analysis of *Figure 2*.

than in control. We then detected the inflammatory cytokine genes which can be regulated by NF-κB signaling pathway in GNAQ-KO decidual cells. As show in *Figure 2*, there is a significant upregulation of interlukin-11 (IL-11) (*Figure 2D*) and interlukin-1β (IL-1β) (*Figure 2E*) in GNAQ-KO decidual cells. Consistently, the IL-1β protein levels were also significantly increased upon Gαq deficiency in decidual cells (*Figure 2F* and *Figure 2—figure supplement 2E*). STAT3 is maximally expressed in decidual cells during late-secretory phase (*Dimitriadis et al., 2006*), phosphorylated STAT3 is the activated form of STAT3 and is the convergent downstream effector of IL-11 (*Taniguchi and Karin, 2014*; *Heichler et al., 2020*) and IL-1β (*Yao et al., 2017*) signaling. We performed western blot to analyze the STAT3 phosphorylation level in decidual cells upon Gαq knockout, and found that Gαq deficiency could significantly enhance the phosphorylation of STAT3 in DSCs (*Figure 2F* and *Figure 2—figure supplement 2D*).

To further investigate whether the over-active NF-κB pathway was implicated in the abnormal decidualization in Gαq knockout decidual cells, we performed rescue experiments using a potent NF-κB inhibitor pyrrolidinedithiocarbamate ammonium (PDTC), which inhibits IκB phosphorylation. The treatment of PDTC inhibitor blocked NF-κB translocate to the nucleus (*Figure 2—figure supplement 1B*), reduced the overexpression of *IL-11*, *IL-1β*, and also the phosphorylated status of STAT3 (*Figure 2G–I* and *Figure 2—figure supplement 2F, G*). Meanwhile, the NF-κB inhibitor slowed down the advanced expression of decidual markers *PRL* and *IGFBP-1* in GNAQ-KO decidual cells (*Figure 2J–L* and *Figure 2—figure supplement 2H*). Collectively, Gαq modulated the expression of inflammatory cytokine by inhibiting over-activation of NF-κB signaling pathway during decidualization.

## Cytoplastic localization of Gαq plays a main role in directing the progression of decidualization by regulating the expression of *NFκBIA*

The above data suggested that Gαq inhibited over-activation of NF-κB signaling pathway to direct the progression of decidualization. It has been reported that inactivation NF-κB is anchored to inhibitor κBα (IκBα) in the cytosol. Upon IκBα phosphorylated by IκB kinase (IKK), it undergoes ubiquitin-dependent degradation (*Maniatis, 1997*), and the 'free' NF-κB then translocates into the nucleus to induces numerous NF-κB targeted inflammatory gene transcriptions. Interestingly, we found that both mRNA and protein level of *NFκBIA* were significantly attenuated in early GNAQ-KO decidual cells, but not in Gαq deficiency proliferative stroma compared with control cells (*Figure 3A, B* and *Figure 3—figure supplement 1A*). However, there is no obvious difference of ratio of phosphorylated IκBα in GNAQ-KO decidual cells (*Figure 3B* and *Figure 3—figure supplement 1B*). To better understood the role of IκBα in Gαq directing the progression of decidualization, we attempted to rescue these abnormalities by supplying the WT IκBα into the Gαq deficiency decidual cells. As expected, nuclear translocation of the NF-κB subunit p65 and p50 were less in *NFκBIA* overexpression GNAQ-KO decidual cells (*Figure 3C*). Subcellular fraction assay further confirmed the above observation (*Figure 3D* and *Figure 3—figure supplement 1C, D*). Meanwhile, overexpression of the IκBα, but not lentivirus transfected with empty vector reduced the phosphorylation of STAT3 (*Figure 3F* and *Figure 3—figure supplement 1F*) and significantly alleviated advanced progression of decidualization in GNAQ-KO decidual cells (*Figure 3E, F* and *Figure 3—figure supplement 1E*).

Interestingly, we detected nuclear localization of Gαq especially in DSCs (*Figure 1—figure supplement 1A, D, E*). In spite of extensive studies on its function located in cytoplasm, little is known for the function of Gαq in the nuclear. To further explore the nuclear role of Gαq, we performed chromatin protein isolation assays and detected Gαq in the nucleoplasm (*Figure 4A*). Next, we evaluated the function of Gαq in different compartments by re-expressing HA-tagged WT Gαq (HA-Gαq) or NES Gαq plasmids contained a fused potent nuclear export sequence into Gαq knockout decidual cells (*Figure 4B*), and found that both of them can restore *NFκBIA* expression (*Figure 4C*), downregulated the expression of *IL-11*, *IL-1β*, and phosphorylation of STAT3 (*Figure 4D–F*), and alleviated the advanced progression of decidualization in Gαq deficiency decidual cells (*Figure 4F*). These results

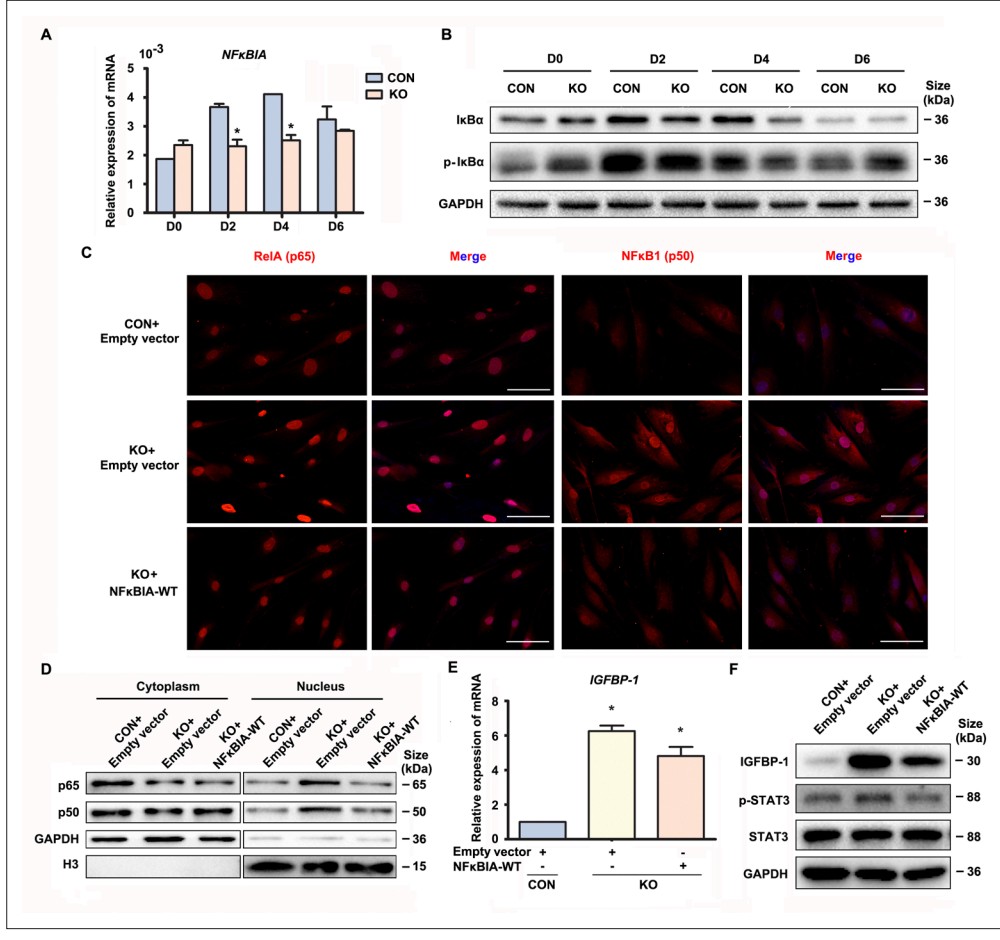

**Figure 3.** Gαq deficiency led to over-activated nuclear factor (NF)-κB signaling pathway during decidualization. (**A**) qRT-PCR analysis of *NFκBIA* mRNA expression in proliferative (D0) and decidualized human endometrial stromal cell (HESC) treated with differential culture medium for 2, 4, and 6 days (*n* = 3). (**B**) Western blot analysis IκBα and p-IκBα expression in whole-cell lysates of proliferative (D0) and decidualized HESC treated as in (**A**). GAPDH was used as the loading control. (**C**) Immunofluorescence staining of p65 and p50 in CON and KO stromal cells and treated with lentivirus transfected with empty vector or *NFκBIA* overexpression lentivirus for 3 days. Scale bars, 100 μm. (**D**) Western blot analysis of the p65 and p50 expression in cytoplasm and nucleus of decidualized HESC treated with lentivirus transfected with empty vector or *NFκBIA* overexpression lentivirus for 3 days. GAPDH and H3 were used as the loading controls. (**E**) qRT-PCR analysis of *IGFBP-1* mRNA expression levels in CON and KO HESC treated as in (**C**) (*n* = 3). (**F**) Western blot analysis of the IGFBP-1, p-STAT3, and STAT3 expression in whole-cell lysates of decidualized HESC treated as in (**C**). GAPDH was used as the loading controls. Representative data are shown from three independent experiments. (**A**) was analyzed with unpaired Student's *t*-test. (**E**) was calculated with one-way analysis of variance (ANOVA) with Bonferroni's multiple comparison tests. *p < 0.05.

The online version of this article includes the following source data and figure supplement(s) for figure 3:

**Source data 1.** Original blot images of *Figure 3*.

**Figure supplement 1.** Relative of western blot analysis of *Figure 3*.

---

suggested that cytoplastic localization of Gαq protein played a dominate role in inhibiting the overexpression of inflammatory cytokine and directing the progression of decidualization by regulating the expression of *NFκBIA*.

## Gαq stimulates PKD/PKCμ activation during decidualization

To investigate the downstream signaling pathways through which Gαq could mediate *NFκBIA* expression to regulate the progression of decidualization, we profiled the Gαq downstream PKC isoforms related signaling pathways by western blots (*Yuan et al., 2000*). Among the more than 10 different PKC isoforms, we found that PKCα/β and PKD/PKCμ were dramatically expressed in

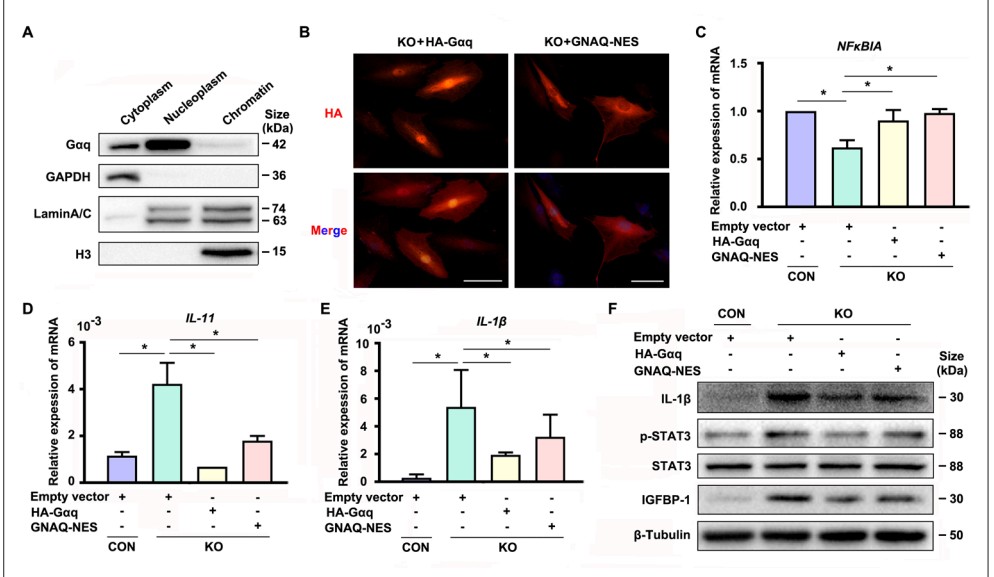

**Figure 4.** Cytoplasmic localized Gαq played the main role in regulating the expression of *NFκBIA* during decidualization. (**A**) Western blot analysis of the Gαq expression in cytoplasm, nucleoplasm, and chromatin of WT human endometrial stromal cell (HESC) treated with differential medium for 4 days. GAPDH, Lamin A/C, and H3 were used as the loading controls for different cellular compartments. (**B**) Immunofluorescence staining of hemagglutinin (HA) in KO differentiated stromal cells infected with lentivirus encoding HA-tagged HA-Gαq or *GNAQ*-NES for 2 days. Scale bars, 100 μm. (**C–E**) qRT-PCR analysis of *NFκBIA, IL-11*, and *IL-1β* mRNA expression levels in CON and KO HESC treated with control HA-Gαq or *GNAQ*-NES overexpression lentivirus for 3 days (*n* = 3). (**F**) Western blot analysis of the IL-1β, p-STAT3, STAT3, and IGFBP-1 expression in whole-cell lysates of CON and KO HESC treated as in (**C**). β-Tubulin was used as the loading controls. Representative data are shown from three to five independent experiments. (**C–E**) were calculated with one-way analysis of variance (ANOVA) with Bonferroni's multiple comparison tests. *p < 0.05.

The online version of this article includes the following source data for figure 4:

**Source data 1.** Original blot images of *Figure 4*.

decidual cells, while PKCθ, δ/θ, ζ / λ were only weakly expressed (data not shown). As shown in *Figure 5*, PKD/PKCμ$^{ser744/748}$ phosphorylation level was significantly reduced (*Figure 5A*), but we detected no significant alteration in PKCα/β phosphorylation in Gαq deficiency differentiated stromal cells (*Figure 5—figure supplement 1A*). To further investigate whether the pathway was implicated in the defective decidualization in GNAQ-KO cells, we performed rescue experiments using the PKC signaling-specific activator Phorbol 12-myristate 13-acetate (PMA) (*Figure 5B, C*). Interestingly, at a concentration of 1 nM, PMA was able to activate PKD/PKCμ pathway and abolished the aberrant expression of decidual marker IGFBP-1 in Gαq knockout cells (*Figure 5B*), but inhibited the activation of p-PKCα/β in normal decidual cells (*Figure 5—figure supplement 1B*). In contrast, PMA had no significant effect on PKA activation at this concentration (*Figure 5—figure supplement 1C*). It has been reported that Gαq activation is sufficient to stimulate sustained PKD activation via PKC (*Yuan et al., 2000*), but we detected no significant alteration in the phosphorylation of PKCα/β (*Figure 5—figure supplements 1A and 2A*), indicating that this PKC activation is not sufficient for PKD phosphorylation in GNAQ-KO decidual cells. The G-protein adenylyl cyclase stimulatory G protein (Gs) couples many receptors to adenylyl cyclase and mediates receptor-dependent adenylyl cyclase activation, resulting in increasing level of intracellular cAMP to activate PKA to regulate the expression of major decidual factors. However, the phosphorylation level of PKA did not respond obviously to Gαq knockout (*Figure 5—figure supplement 1D*). Taken together, these results implied that Gαq-mediated HESC decidualization mainly by stimulating the PKD/PKCμ activation.

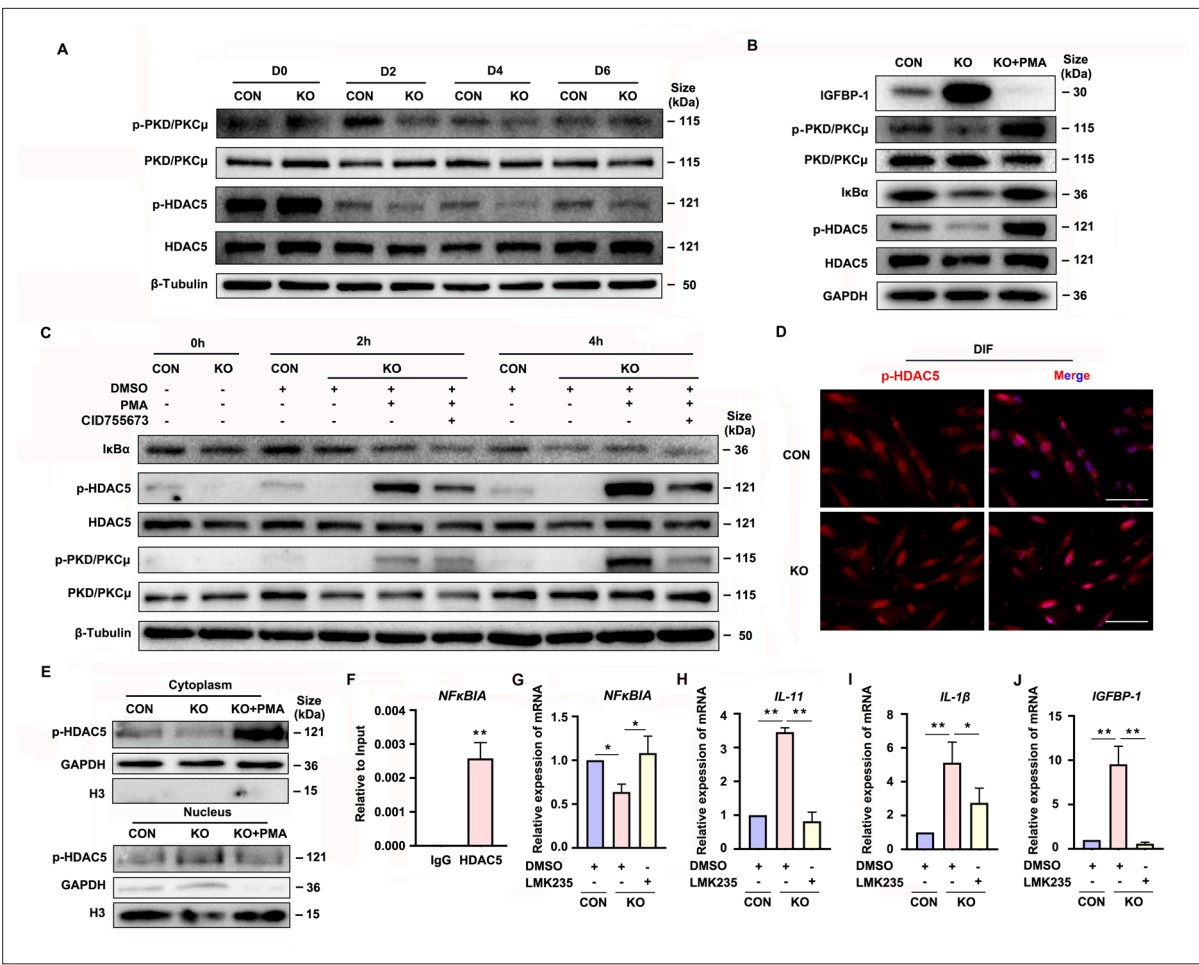

**Figure 5.** Gαq deficiency attenuated PKD/PKCμ activation and reduced HDAC 5 phosphorylation level to inhibit *NFκBIA* expression during decidualization. (**A**) Western blot analysis of the p-PKD/PKCμ, PKD/PKCμ, p-HDAC5, and HDAC5 expression in whole-cell lysates of proliferative (D0) and decidualized human endometrial stromal cell (HESC) treated with differential culture medium for 2, 4, and 6 days. β-Tubulin was used as the loading controls. (**B**) Western blot analysis of the IGFBP-1, p-PKD/PKCμ, PKD/PKCμ, I κ Bα, p-HDAC5, and HDAC5 expression in whole-cell lysates after treatment with PKD/PKCμ activator, Phorbol 12-myristate 13-acetate (PMA, 1 nM) in CON and KO HESC treated with differential culture medium for 4 days. GAPDH was used as the loading controls. (**C**) Western blot analysis of the I κ Bα, p-HDAC5, HDAC5, p-PKD/PKCμ, and PKD/PKCμ expression in whole-cell lysates after treatment with PMA (1 nM) and or PKD/PKCμ inhibitor CID755673 for 0, 2, and 4 hr in CON and KO HESC treated with differential culture medium for 4 days. β-Tubulin was used as the loading controls. (**D**) Immunofluorescence staining of p-HDAC5 in CON and KO decidualized stromal cells treated with differential culture medium for 4 days. Scale bars, 100 μm. (**E**) Western blot analysis of the p-HDAC5 expression in cytoplasm and nucleus after treatment with PMA (1 nM) in CON and KO HESC treated with differential culture medium for 4 days. GAPDH and H3 were used as the loading controls. (**F**) Chromatin immunoprecipitation (ChIP)-qRT-PCR analysis of enrichment of HDAC5 on promoters of *NFκBIA* in WT HESC treated with differential culture medium for 4 days (*n* = 3). (**G–J**) qRT-PCR analysis of *NFκBIA*, *IL-11*, *IL-1β*, and *IGFBP-1* mRNA expression levels after treatment with HDAC5 inhibitor LMK235 (2 μM), in CON and KO HESC treated with differential culture medium for 4 days. Representative data are shown from three independent experiments (*n* = 3). (**F**) was analyzed with unpaired Student's *t*-test. (**G–J**) were calculated with one-way analysis of variance (ANOVA) with Bonferroni's multiple comparison tests. *p < 0.05 and **p < 0.01.

The online version of this article includes the following source data and figure supplement(s) for figure 5:

**Source data 1.** Original blot images of *Figure 5*.

**Figure supplement 1.** Gαq deficiency has no significant affection in PKCα/β and PKA C phosphorylation during decidualization.

**Figure supplement 1—source data 1.** Original blot images of *Figure 5—figure supplement 1*.

**Figure supplement 2.** Relative of western blot analysis of *Figure 5—figure supplement 1A*.

## PKD/PKCμ stimulates the *NFκBIA* expression in decidual cells via inducing HDAC5 phosphorylation

Gene transcription can be regulated by the transcriptional factor and epigenetic mechanism (*Lee et al., 2019*). As one of the epigenetic modifications, the histone acetylation can induce the target gene transcription. It has been reported that PKD/PKCμ could directly phosphorylated HDAC5, one of the deacetylases (*Vega et al., 2004*; *Cho et al., 2015*). To determine the potential mechanism of PKD/PKCμ signaling pathway in regulating the *NFκBIA* transcription during decidualization, we investigated the phosphorylation level of HDAC5 in GNAQ-KO HESC. As shown in *Figure 5A–C*, the HDAC5 phosphorylation level was significantly reduced in GNAQ-KO decidual cells. Then, the change was further validated by using the PKD/PKCμ activator and inhibitor during decidualization in GNAQ-KO cells. As expected, the reduced phosphorylation level of HDAC5 upon Gαq knockout was dramatically rescued after activator PMA treatment (*Figure 5B, C*), and co-treatment with PKD/PKCμ inhibitor, CID755673 blocked the PMA-induced HDAC5 phosphorylation (*Figure 5C*). These results suggested that PKD/PKCμ regulated the HDAC5 phosphorylation status during decidualization.

Phosphorylated HDAC5 could stimulate its nuclear export (*Weeks et al., 2017*; *Pietruczuk et al., 2019*), and HDAC5 nuclear export enhances histone acetylation to activate the target gene expression (*Cho et al., 2015*). We postulated that the attenuated *NFκBIA* expression may due to reduced histone acetylation on its promotor region caused by the abnormal nuclear localized HDAC5 when its phosphorylation was reduced. To prove this assumption, firstly, it showed that evidently decreased HDAC5 phosphorylation level in cytoplasm of GNAQ-KO decidual cells compared with the control decidual cells (*Figure 5D, E*), and PKD/PKCμ activator, PMA, dramatically reversed the reduced phosphorylation level of HDAC5 in cytoplasm (*Figure 5E*), further corroborated that the phosphorylated HDAC5 could stimulate its nuclear export. We then investigated whether HDAC5 bind the promoter and repressed its transcription. Chromatin immunoprecipitation (ChIP) assay verified the enrichment of HDAC5 in *NFκBIA* promotor (*Figure 5F*). Further study detected *NFκBIA* expression in GNAQ-KO decidual cells treated with HDAC5 inhibitor, LMK235. As shown in *Figure 5G*, LMK235 significantly rescued the *NFκBIA* expression in GNAQ-KO decidual cells, downregulated its effectors IL-11 (*Figure 5H*) and IL-1β (*Figure 5I*) expression and alleviated the advanced progression of decidualization (*Figure 5J*) in GNAQ-KO decidual cells. Moreover, to further confirm this assumption, we also detected IκBα expression in PMA or CID755673-treated GNAQ-KO decidual cells. It was found that IκBα expression significantly increased in the PMA-treated GNAQ-KO decidual cells (*Figure 5B, C*), and CID755673 blocked the IκBα expression (*Figure 5C*). Collectively, these results clearly demonstrated that PKD/PKCμ signaling pathway regulated the *NFκBIA* expression by regulating HDAC5 phosphorylation level and nuclear export during decidualization.

## Human blastocyst's hatching is deferred and embryo adhesion is failed when co-culture with GNAQ-KO decidua cell

Both pro-inflammatory and anti-inflammatory cytokine (of endometrial and embryonic origin) are potentially required to maintain normal pre-implantation embryo development and blastocyst hatching, leading to successful implantation (*Pathak et al., 2021*). Deferred or failed blastocyst hatching can lead to blastocyst implantation failure (*Khorram et al., 2000*), and delayed or failed blastocyst adhesion miss the 'implantation window' lead to infertility or early pregnancy loss. Considering this case, we tested the outcome of over-active inflammation in GNAQ-KO decidua on normal blastocyst hatching and adhesion by performing a co-culture assay. Results show that, the time required for initiated hatching in the CON group was about 8 hr after co-culture (22.2%, 6/27), whereas the initiation of human blastocyst hatching deferred in the GNAQ-KO (3.7%, 1/27) (*Figure 6A–C*). The speed and areas of human blastocyst outgrowth in CON group were faster and larger than in GNAQ-KO group at the same co-culture time (*Figure 6A, B*), corresponding the adhesion rate was notably higher in the CON than in the GNAQ-KO group (29.6%, 8/27 and 11.1%, 3/27, respectively) after co-culture for 72 hr (*Figure 6A–C*). All of these suggested that the over-active inflammation status of GNAQ-KO decidua led to the deferred human blastocyst initiation hatching, arrested outgrowth, and/or failed adhesion.

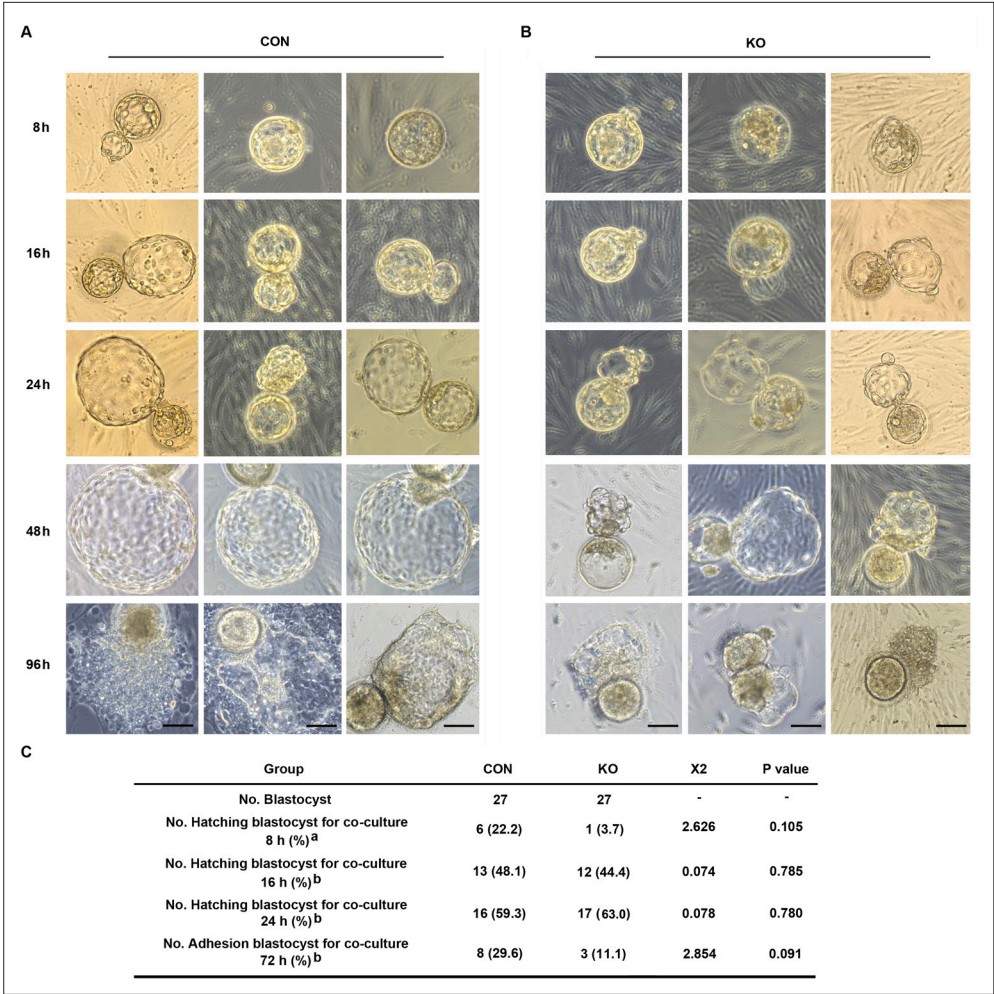

**Figure 6.** Gαq deficiency in decidual cells inhibits the blastocyst hatch and adhesion in the embryo-endometrial decidual cells co-culture assay. (**A, B**) The process of blastocysts hatching, expansion, and adhesion progressively with outgrowth into the stromal cell layer. Images captured from typical day 5–6 blastocysts co-cultured on a layer of CON or KO stromal cells treated with differential culture medium. Scale bar, 100 µm. (**C**) Table of statistical analyses for the role of stromal Gαq for blastocyst hatching and adhesion in the co-culture assay. $n = 5$ independent experiments. (**C**) analyzed with correction Chi-square test (a) and Chi-square test (b).

The online version of this article includes the following source data for figure 6:

**Source data 1.** Table of statistical analyses for the role of stromal Gαq for blastocyst hatching and adhesion in the co-culture assay.

## Endometrial Gαq expression is aberrantly decreased in RPL women

Furthermore, we evaluated Gαq expression in the decidua tissues of women with RPL during the first trimester. By comparing the Gαq mRNA and protein levels in decidua tissue between 13 women with normal pregnancy and 13 with RPL, we found that the endometrial Gαq level was significantly lower in RPL samples than in normal samples (*Figure 7A–C*), *IL-11*, *IL-1β* mRNA, and protein or *IL-6* mRNA expression were significantly higher in the RPL samples than in the normal samples during the first trimester (*Figure 7D–I*), consistent with our in vitro studies. Our novel findings collectively suggested that abnormal endometrial Gαq expression contribute to RPL.

## Discussion

Embryo implantation involves a complex series of events that establishes the connection between maternal and embryonic tissues (*Sharma et al., 2016*). Decidualization is characterized by a biphasic

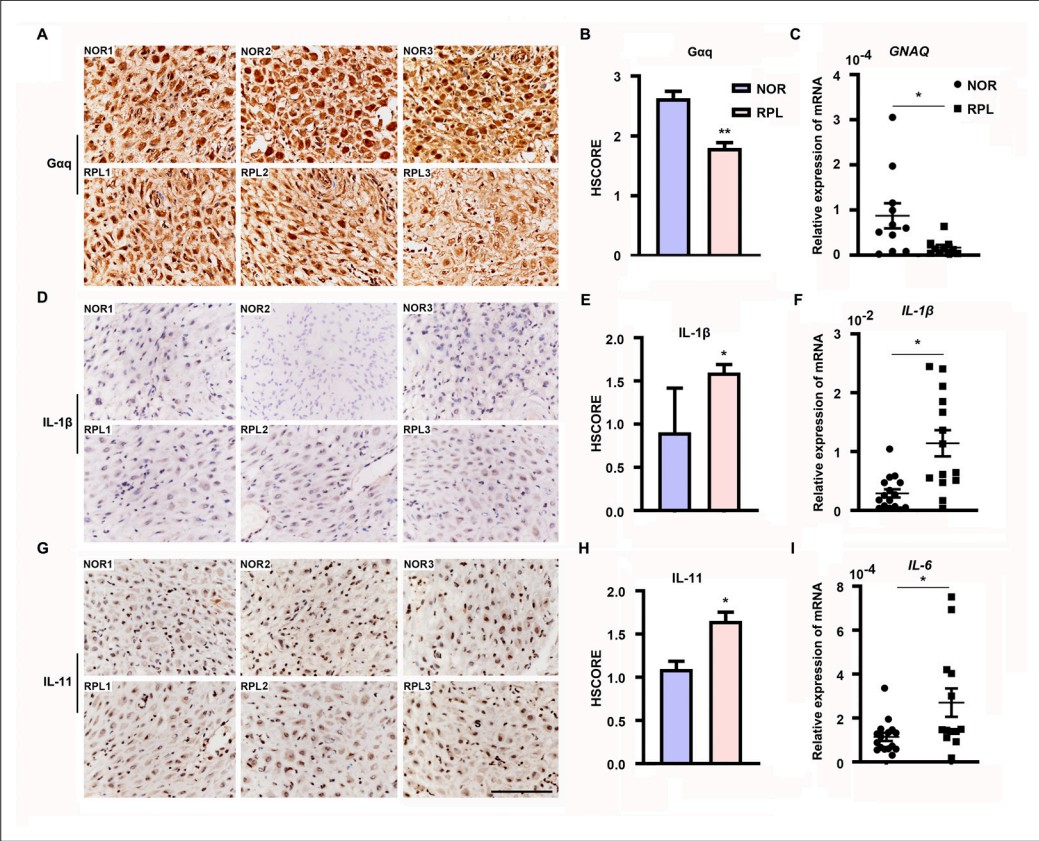

**Figure 7.** Expression of decidual Gαq is decreased in women with recurrent pregnancy loss (RPL). (**A, D, G**) Immunohistochemical staining of Gαq, IL-1β, and IL-11 in the deciduas of women with normal early pregnancies (*n* = 13) and those with RPL (*n* = 13). Scale bar, 100 μm. S, stroma cells; NOR, normal. (**B, E, H**) HSCORE of immunostaining for Gαq, IL-1β, and IL-11 in decidual cells. (**C, F, I**) qRT-PCR analysis of *GNAQ*, *IL-1β*, and *IL-6* mRNA expression in the deciduas of women with normal early pregnancies and women with RPL. Representative data were analyzed with unpaired Student's *t*-test. *p < 0.05 and **p < 0.01.

gene expression dynamic that likely corresponds to different phases in the establishment of the fetal–maternal interface (*Rytkönen et al., 2019*). Secretions from decidualized ESCs promote invasion of trophoblast cells by activating proteases and altering expression of adhesion-related molecules (*Vinketova et al., 2016*). The decidual secretions contain high amounts inflammatory cytokine that include IL-1β, IL-6, and IL-11 (*White et al., 2007*; *Sharma et al., 2016*). Both pro-inflammatory and anti-inflammatory cytokine are potentially required to maintain normal pre-implantation embryo development and blastocyst hatching, leading to successful implantation (*Pathak et al., 2021*). But, excessively elevated levels of inflammatory cytokine likely induce ER stress in the invading EVTs (*Lee et al., 2019*). Above perspectives on blastocyst hatching and trophoblast have immense implications in the clinical management of human infertility. However, few functionally and indispensable factors could hold the key as master regulators of inflammatory cytokine in the context of decidual cells. In the present study, we uncovered that Gαq play an important role in balancing the cytokines expression and decidual progression in response to deciduogenic cues.

Transcriptional patterns suggested that the early decidualization inflammatory state is dominated by STAT signaling. In the late phase, however, STATs are downregulated and the transcription of NF-κB repressors decreases, possibly leading to NF-κB pathway activation (*Rytkönen et al., 2019*). In this study, we showed that, Gαq deficiency attenuated *NFκBIA* mRNA expression in early decidual cells (D2 and D4), but not late decidual cells and proliferative stromal cells. Meanwhile, we detected the expression of *NFκBIA* showed an early upregulated (D3) and late downregulated (D6) pattern in control decidual cells which consistent with Wagner's study (*Rytkönen et al., 2019*), and the phosphorylation level of IκBα with the same pattern of *NFκBIA*. These expression and phosphorylation

patterns of inhibitor κBα suggest that the early decidualization inflammatory state is also balanced by Gαq inhibition for NF-κB signaling.

In addition, we detected nuclear localization of Gαq protein in DSCs. Previous studies have shown that G proteins, and their downstream effector that are found in the nucleus (**Spiegelberg and Hamm, 2005**; **Campden et al., 2015**). There is a substantial body of evidence documenting other membrane proteins, for example, epidermal growth factor receptor (EGFR) family members and PD-L1 to be localized in the nucleus (**Wang et al., 2010b**, **Wang et al., 2010c**, **Yu et al., 2020**). The translocation of EGFR family members involves the recognition of the nuclear localization signal (NLS) by the nuclear transporter importin proteins, thus directing the proteins transport through the nuclear pore complex to the nucleus (**Hsu and Hung, 2007**; **Wang et al., 2010a**; **Wang et al., 2012**). Unlike EGFR or PD-L1, Gαq does not contain a classical NLS (data not shown) (**Yu et al., 2020**). Even with a clear nuclear localization, in this study, we found that cytoplastic localization of Gαq played a dominate role in regulating the expression of *NFκBIA* to balance the progression of decidualization. Further study is needed to explore the potential roles and mechanism of nuclear localized Gαq during decidualization.

We also investigated the signaling pathways through which Gαq could mediate its differentiation effects in HESC. It is well known that Gαq directly activates PLC β, which generates the second messenger DAG. DAG activates PKC by binding their C1 domains (**Chen et al., 2017**) and DAG induces the activation of PKD/PKCμ by PKC-dependent phosphorylation (**Wang, 2006**). PMA as a PKC agonist, at the concentration of 100 nM was able to mimics the DAG to activate PKC. Interestingly, we found that PMA at the low concentration of 1 nM can activate PKD/PKCμ but not the PKCα/β or PKA in Gαq deficiency and normal decidual cells. These findings revealed that Gαq stimulate PKD/PKCμ activation may not via protein kinase C during decidualization, and suggest that there may be PKC-independent pathways stimulated by Gαq that are responsible for PKD/PKCμ activation during decidualization. Thus, further study is warranted to explore further alternative mechanism of Gαq involved in the modulation of PKD/PKCμ activation during decidualization.

Previous studies have investigated that HDACs universally expressed in ESCs and decidual cells (**Pavličev et al., 2017**) which play an important role during decidualization (**Sakai et al., 2003**). However, just HDAC4 and HDAC5 expression have a dramatic change during the transformation of the endometrial stromal fibroblasts into secretory decidual cells (**Pavličev et al., 2017**). While, in our present work, HDAC4 was only weakly expressed (data not shown). HDAC5 belongs to the class IIa HDAC family and phosphorylated form has been known to shuttle between the cytosol and the nucleus. HDAC5 phosphorylation can be regulated by PKD/PKCμ or PKA (**Cho et al., 2015**; **Weeks et al., 2017**; **Pietruczuk et al., 2019**), which would promote HDAC5 shuttling to the cytosol

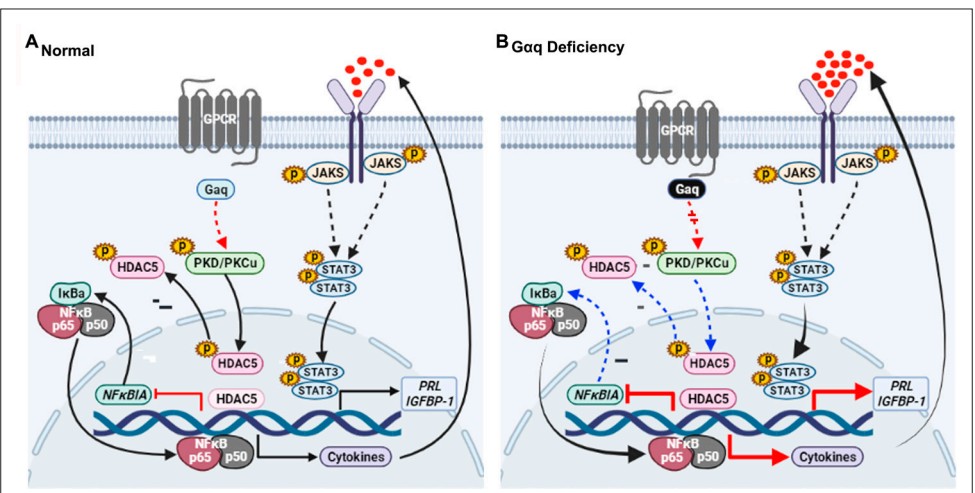

**Figure 8.** Model outlining the functions of Gαq during decidualization. (**A**) During normal decidualization, Gαq-PKD/PKCμ signal inactive HDAC5 through phosphorylation to induce the I κ B expression and limit the nuclear factor (NF- κ B-mediated inflammatory response for endometrial decidual homeostasis). (**B**) In Gαq deficiency decidual cells, Gαq-PKD/PKCμ signal is defective, and excessive nuclear location of HDAC5 reduce the I κ B expression and over-active NF- κ B pathway-mediated inflammatory response result in enhanced inflammatory decidualization.

and relieve its transcription suppress function. Our results demonstrate that PKD/PKCμ can serve as a HDAC5 kinase downstream of Gαq activation in decidual cells. Based on these observations, it suggested that the nuclear HDAC5 activity is precisely modulated both at the protein level and post-translational level through phosphorylation.

In conclusion, our results proved that Gαq expression is required for decidualization during early pregnancy and that a decrease in Gαq expression leads to over-activated inflammatory and aberrantly advanced decidualization progression (*Figure 8A, B*). The results also provided new insights into the mechanism underlying the action of Gαq to balance the cytokines expression and decidual procedure in response to deciduogenic cues and confirm its important contribution in HESC decidualization. Meanwhile, Gαq-deficient decidualization deferred or even destroyed blastocyst hatching and adhesion in vitro. In addition, we found that Gαq expression was significantly lower in RPL samples than in normal endometrial samples, indicating the significance of Gαq to decidualization in vivo. Based on these finding, we demonstrated that, for a successful early pregnancy, a proper decidualization with homeostasis of cytokines secretome govern by Gαq is pre-requested, which provided a new hint for understanding the early pregnancy loss.

## Materials and methods

### Human endometrial tissue samples

This study was approved by the Medical Ethics Committee of The First Affiliated Hospital of Xiamen University, The Xiamen Maternal and Child Health Hospital, and The Third Affiliated Hospital of Guangzhou Medical University, and written informed consent was obtained from all participants in accordance with the guidelines in The Declaration of Helsinki 2000. Informed consents were obtained for all the subjects before their inclusion in the study. Endometrial tissue subjects of menstrual cycles were collected from women who visited the Department of Obstetrics and Gynecology, Reproductive Medical Center, The Third Affiliated Hospital of Guangzhou Medical University from January to October 2020. None of the subjects was on hormonal treatments for at least 3 months prior to the procedure. Normal subjects with regular cycles (ranging from 25 to 35 days) were undergone endometrial sampling on a specific cycle day under a protocol approved by the The Third Affiliated Hospital of Guangzhou Medical University. The control group consisted of randomly selected women who underwent legal termination of an apparently normal early pregnancy at the same hospital during the same period, without medical reasons, history of pregnancy loss or any other pregnancy complication. The study cohort included women with normal fetal chromosome confirmed RPL with two or more consecutive pregnancy losses of undetermined etiology, characteristics of these patients were listed in *Supplementary file 1A*. Some proportions of tissue samples were immediately snap frozen in liquid nitrogen for subsequent RNA analysis and some were formalin fixed, embedded in paraffin, and sectioned for immunohistochemistry.

### HESC culture and in vitro decidualization

Immortalized HESC line was purchased from the American Type Culture Collection (ATCC CRL-4003) and cultured according to the manufacturer's instructions (*Krikun et al., 2004*). The identity of the cell lines has been authenticated by ATCC, and we have confirmed that the cell lines tested negative for mycoplasma contamination. Briefly, HESC was cultured in Dulbecco's modified Eagle's medium (DMEM)/F12 (Gibco) supplemented with 1% penicillin and 1% streptomycin, 10% charcoal-stripped fetal bovine serum (CS-FBS, Biological Industries), 3.1 g/l glucose, 1 mM sodium pyruvate, 1% Insulin-Transferrin-Selenium (ITS) (Gibco), and 500 ng/ml puromycin at 37°C in a humidified chamber with 5% $CO_2$. To induce decidualization in vitro, HESC was treated with differential medium (DMEM/F12 with 2% CS-FBS) containing 1 μM medroxyprogesterone 17-acetate (MPA, Sigma, M1629), and 0.5 mM dibutyryl cAMP (dbcAMP, Sigma, D0627). The medium was changed every 48 hr. Other reagents including inhibitors, activators are listed in *Supplementary file 1B*.

### Primary human endometrial cell culture

Primary HESC culture was performed as described previously (*Jiang et al., 2020*). In brief, endometrial tissues from women with regular menstrual cycles were collected, washed, and then minced into <1-$mm^3$ pieces. After enzymatic digestion primary ESCs were filtered through a 250-mm mesh sieve,

cultured in phenol red free DMEM at 37°C in a humidified chamber with 5% CO2. The ESCs were directly seeded, grown to confluence and then passaged for continuous cell culture. The purity of stromal cells was verified by immunofluorescence staining with antibody against the stromal-specific vimentin (1:200, Cat# ab92547, Abcam, USA) at the third passage.

## Generation and validation of GANQ-KO cells

The GNAQ knockout (GNAQ-KO) cells were constructed using the CRISPR/Cas9 gene-editing system. The CRISPR plasmid contains expression cassettes of Cas9, chimeric gRNA, and CopGFP. The target sequences of gRNAs were designed using the MIT online tool (http://crispr.mit.edu/), and gRNAs were designed to knock out GNAQ: 5'-TTATTGTGCTCATACTTGTA-3'. The CRISPR plasmids were transfected into HEK293T cells using JET-PEI (Polyplus-transfection), with lentivirus genomic plasmids for lentiviral packaging. The lentivirus used to infect HESC and single-cell colonies of GFP-positive cells were picked, and genomic DNA was extracted from the GFP-positive cells and amplified with primers located ~300 bp from the target site. PCR products were Sanger sequenced, and sequences were aligned using the Beacon Designer (PREMIER Biosoft, USA) sequence alignment tool. The GNAQ knockout HESC cells were further validated the Gαq protein expression level by western blot.

## RNA-seq and bioinformatics analysis

Three biological replicates for control and GNAQ-KO HESC with differential medium treatment for 3 days, treatment cells were collected in 1 ml TRIzol for a 9.6-cm dish. RNA was extracted and followed by the TRIzol Reagent kit from Invitrogen. RNA-seq and bioinformatics analysis were performed at the Beijing Genomics Institute (BGI). RNA-seq was carried out at a depth of 15–20 million single-end reads. Libraries for sequencing were generated from the double-stranded cDNA using the DNBSEQ manufacturer's protocol. Library quality control was checked using the Agilent 2100 Bioanalyzer. Sequencing was performed on a Combined probe anchored polymerization technology (cPAS). The differentially expressed genes were identified on the basis of multiple test corrected p-value and fold change. Genes were considered as being significantly differentially expressed if the p-value was <0.05 and absolute fold change >2.

## Immunohistochemistry and immunofluorescence

Endometrial tissues from RPL subjects and various staged controls were examined. Immunostaining was performed as described previously (Jiang et al., 2020). The endometrial tissues were fixed in 4% paraformaldehyde (Sigma) and then embedded in paraffin. After deparaffinization, rehydration, antigen retrieval, inhibition of endogenous peroxidase activity, and blocking, sections were incubated with rabbit polyclonal antibody against Gαq (Abcam, Cat# ab75825, 1:200) overnight. Immunostaining was performed using peroxidase-labeled Avidin–Biotin system. For confocal microscopy, control and GNAQ-KO HESC cultured on glass chamber slides. HESC was fixed in methanol for 5 min, permeabilized with 0.1% Triton X-100 in phosphate-buffered saline (PBS) for 10 min and blocked in 0.5% bovine serum albumin in PBS for 1 hr. Endogenous proteins were stained with rabbit polyclonal antibody against Gαq, rabbit monoclonal antibody against Ki67 (Abways Technology, Cat# CY5542, 1:200), rabbit polyclonal antibody against p50 (Abcam Cat# ab32360, 1:200), rabbit polyclonal antibody against p65 (Abcam, Cat# ab32536, 1:200), and rabbit monoclonal antibody against HA-Tag (Cell Signaling Technology, Cat# 3724, 1:200) overnight at 4°C. Fluorescence (cyanine3)-conjugated secondary antibodies were used to visualize the signal and nuclei were stained with DAPI (Sigma, 1 µg/ml), respectively.

## ChIP assay and RT-qPCR

ChIP assay was performed as described previously (Jiang et al., 2016). In brief, the normal differentiated stromal cells were treated with 1% formaldehyde (CST, 12606S)-PBS solution to crosslink the protein–DNA complex. Chromatin was sheared by sonication to obtain the DNA fragment and the sheared chromatin was precleared by 15 µl protein G Dynabeads (Invitrogen, 10007D). Chromatin fragments were incubated with 5 µg anti-HDAC5 antibodies (Cell Signaling Technology, Cat# 98329, 1:50) overnight at 4°C. Normal mouse IgG (Santa Cruz Biotechnology, SC-2025) was used as a negative control for nonspecific immunoprecipitation. The chromatin–antibody complex was pull-downed by the protein G Dynabeads after rigid washing. The beads were suspended in elution buffer and the

precipitated protein/DNA complexes were eluted from the antibodies/beads. The resulting protein/DNA complexes were subjected to cross-link reversal and the captured DNA was purified by phenol/chloroform extraction and ethanol precipitation. Specific primers were used to detect immunoprecipitated chromatin fragments, as well as input chromatin (primers for ChIP assay are listed in *Supplementary file 1C*).

## RNA extraction and quantitative polymerase chain reaction (qRT-PCR)

Total RNA was extracted using the TRIzol RNA purification kit (Invitrogen), and 1 µg of total RNA was used to synthesize cDNA using the Takara PrimeScript RT reagent kit according to the manufacturer's instructions. Expression levels of different genes were determined by qRT-PCR using SYBR Green Supermix (Roche, Germany), using the ABI Q5 Real Time PCR System (Applied Biosystems, USA). All primers for qRT-PCR are listed in *Supplementary file 1C*. The mRNA expression level at given time point or under a given condition was quantified as normalization with respect to the internal control *GAPDH* (mean ± standard error of the mean [SEM]). All measurements were performed at least three times.

## Western blot analysis

Whole-cell extracts were prepared from HESC and subjected to western blot analysis as described previously (*Jiang et al., 2016*). Briefly, cells were lysed with the radioimmune precipitation assay buffer containing a cocktail of protease inhibitors (Roche). After removal of the cell debris, 15–30 µg of the protein extract were loaded onto sodium dodecyl sulfate–polyacrylamide gel electrophoresis (SDS–PAGE) and transferred to polyvinylidene fluoride membrane. The membrane was blocked in 5% skimmed milk (BD) and probed with rabbit polyclonal antibody against Gαq, rabbit monoclonal antibody against IGFBP-1 (Abcam, Cat# ab181141, 1:1000), mouse monoclonal antibody against IL-1β (Origene Inc, Cat# TA506443, 1:1000), rabbit polyclonal antibody against p65 (Abcam, Cat# ab32536, 1:1000), rabbit polyclonal antibody against p50 (Abcam Cat# ab32360, 1:1000), p-STAT3 (Abcam Cat# ab76315, 1:1000), STAT3 (Abcam Cat# ab68153, 1:1000), rabbit polyclonal antibody against IκBa (Abcam Cat# ab32518, 1:1000), rabbit polyclonal antibody against p-NFκB p65(Ser536) (Bioss, Cat# bs-0982R, 1:1000), mouse monoclonal antibody against p-IκB a (Ser32) (Bioss, Cat# bsm-52169R, 1:1000), rabbit monoclonal antibody against p-PKA C(Thr197) (Cell Signaling Technology, Cat# 5661, 1:1000), rabbit monoclonal antibody against PKA C- a (Cell Signaling Technology, Cat# 5842, 1:1000), rabbit polyclonal antibody against p-PKC a /βⅡ (Thr638/641) (Cell Signaling Technology, Cat# 9375, 1:1000), rabbit polyclonal antibody against PKC a (Cell Signaling Technology, Cat# 2056, 1:1000) and p-PKC Antibody Sampler Kit (Cell Signaling Technology, Cat# 9921, 1:1000), mouse monoclonal antibody against HDAC5 (Cell Signaling Technology, Cat# 98329, 1:1000), and rabbit polyclonal antibody against p-HDAC5 (ABclonal, Cat# AP0202, 1:1000). The blots were then incubated with peroxidase-conjugated secondary antibodies, and the signal was detected by chemiluminescence. For the loading control, the membrane was probed with rabbit polyclonal antibody against GAPDH (Abmart, Cat# M20006, 1:5000), mouse monoclonal antibody against β-Tubulin (ABclonal, Cat# AC021, 1:2000), rabbit polyclonal antibody against LaminA/C (ABclonal, Cat# A0249, 1:1000), and rabbit monoclonal antibody against H3 (Abmart, Cat# P30266,1:2000).

## Overexpression vectors and transfection

The HA-tagged (HA sequence: AGCGTAATCTGGAACATCGTATGGGTA) GNAQ or with nuclear export sequence (NES sequence: TTAGCATTAAARTTAGCNGGNTTAGATATCGGNAGC) on C terminal of GNAQ coding sequence (CDS) and HA-tagged NFKBIA was amplified by PCR and cloned into the pLVX-IRES-ZsGreen (Takara, Catalog No. 632187) lentiviral plasmids. The plasmids were then co-transfected into HEK293T cells with lentivirus genomic plasmids for lentiviral packaging. All clones were confirmed by DNA sequencing and analyzed for mRNA and protein expression to confirm overexpression efficiency.

## Nuclear protein extracts

To prepare nuclear extracts, as described previously (*Spiegelberg and Hamm, 2005*; *Yao et al., 2015*). Briefly, the cells were washed three times with PBS and lysed with Buffer A (10 mM Hydroxyethyl piperazine ethanesulfonic acid (HEPES), pH 7.9, 10 mM KCl, 0.1 mM Ethylenediaminetetraacetic acid

(EDTA), 0.1 mM Ethylene glycol tetraacetic acid (EGTA), and 0.15% NP-40) containing 1% protease inhibitor on ice for 30 min (shaking the tube every 10 min). The homogenates were centrifuged at 1500 × g for 5 min at 4°C, and the supernatants (the cytosolic fraction) were discarded. The precipitates were washed three to five times with Buffer A and then resuspended in Buffer B (1 M Tris–HCl pH 7.4, 5 M NaCl, 0.5 M EDTA, 10% Na-deoxycholate, 10% SDS, and 10% Triton X-100) containing 1% protease inhibitor for 30 min (shaking the tube every 10 min). Cellular debris was removed by centrifugation at 12,000 × g for 30 min at 4°C, and the supernatant (the nuclear fraction) was stored at −80°C until further analysis.

### Chromatin protein isolation

To isolate chromatin, as described previously (*Spiegelberg and Hamm, 2005*; *Wu et al., 2014*). Briefly, cells were resuspended with 1 ml Buffer A (10 mM HEPES, pH 7.9, 10 mM KCl, 0.1 mM EDTA, 0.1 mM EGTA, and 0.15% NP-40,) containing 1% protease inhibitor on ice for 30 min (shaking the tube every 10 min). The homogenates were centrifuged at 1500 × g for 5 min at 4°C, and collected the supernatant (P1). Nuclei were washed four times with Buffer A, centrifuged again under the same conditions, and then resuspended with 200 µl Buffer B (50 mM Tris–HCl pH 7.4, 150 mM NaCl, 1 mM EDTA, 0.25% Na-Deoxycholale, 1% NP-40, protease inhibitors as described above) and on ice for 30 min (shaking the tube every 10 min) to release chromatin. Nucleus proteins (P2) and insoluble chromatin-bound proteins were collected by low speed centrifugation (4 min, 1700 × g, 4°C), washed three times in Buffer B, and centrifuged again under the same conditions. The final chromatin-bound proteins pellet (P3) was resuspended in 50 µl Buffer C (1 M Tris–HCl pH 7.4, 5 M NaCl, 0.5 M EDTA, 10% Na-Deoxycholale, 10% SDS, 10% Triton X-100, protease inhibitors as described above). Cellular debris was removed by centrifugation at 20,000 × g for 30 min at 4°C. A quantity of 15 µg cytoplasm was loaded, while 50 µl nucleoplasm or 50 µl chromatin were loaded without being measured.

### Human blastocyst – ESC co-culture

All donated abandoned blastocysts were collected after the consent and approval of the patients based on the guidelines of the ethical committee of The Xiamen Maternal and Child Health Hospital and in accordance with the regulations and guidelines of the People's Republic of China on the ethical principles of the Human Assisted Reproductive Technology and the Helsinki declaration (IRB approval number: KY-2022-028-K01). All donors were informed that their abandoned blastocysts would be used for basic research and that the donation would not affect their clinical treatment. Twelve-well dishes (Thermo Fisher Scientific) were pre-coated with CON and GNAQ-KO stromal cells, to induce decidualization with differential culture medium for 48 hr and changed the medium before co-culture. Unassisted hatching day – 5 or 6 human blastocysts were randomly and individually placed into wells and co-cultured for 96 hr at 37°C with 5% $CO_2$. Images were recorded after 8 hr co-culture. Human blastocyst hatching was defined as the beginning escaped from the zona pellucida. When trophoblast cells grown outward from the blastocysts and the trophoblast cells became visible, these embryos were designated as adhesion-initiating blastocysts. The proportions of hatched and adhered blastocysts, outgrowth area, and outgrowth speed were examined to estimate the implantation capacity of blastocysts in vitro.

### Statistical analysis

SPSS 22.0 software was used for the statistical analysis. The data represented as percentage were evaluated by Chi-square statistics. For the variables expressed as mean ± SEM of at least three independent experiments, statistical significance was evaluated by unpaired *t*-test for normally distributed data two-tailed Student's *t*-tests or one-way analysis of variance, and Mann–Whitney *U*-test for data that did not conform to normal distribution. $p < 0.05$ was considered statistically significance.

## Acknowledgements

We thank Dr Jiahuai Han for kindly providing plasmid encoding NFKBIA, and The First Affiliated Hospital of Xiamen University, The Xiamen Maternal and Child Health Hospital, and The Third Affiliated Hospital of Guangzhou Medical University for their assistance with human sample collection. We

would also like to thank all of the participants for donating samples. This work was supported in part by the National Natural Science Foundation of China 81971536 (to GS), 82288102 (to HW), 81971388 (to SK), 81701457 (to YJ), and 82101726 (to YZ).

## Additional information

### Funding

| Funder | Grant reference number | Author |
| --- | --- | --- |
| National Natural Science Foundation of China | 81971536 | Guixiu Shi |
| National Natural Science Foundation of China | 82288102 | Haibin Wang |
| National Natural Science Foundation of China | 81971388 | Shuangbo Kong |
| National Natural Science Foundation of China | 81701457 | Yufei Jiang |
| National Natural Science Foundation of China | 82101726 | Ying Zhang |

The funders had no role in study design, data collection, and interpretation, or the decision to submit the work for publication.

### Author contributions

Yufei Jiang, Data curation, Funding acquisition, Investigation, Methodology, Writing – original draft, Writing – review and editing; Yan He, Songting Liu, Jinxiang Wu, Data curation, Methodology, Writing – review and editing; Gaizhen Li, Data curation, Investigation, Methodology, Writing – review and editing; Dunjin Chen, Ping Li, Resources, Project administration, Writing – review and editing; Wenbo Deng, Methodology, Project administration, Writing – review and editing; Ying Zhang, Data curation, Funding acquisition, Methodology, Writing – review and editing; Jianing Li, Resources, Writing – review and editing; Longmei Wang, Formal analysis, Methodology, Writing – review and editing; Jiajing Lin, Methodology, Writing – review and editing; Haibin Wang, Resources, Funding acquisition, Project administration, Writing – review and editing; Shuangbo Kong, Funding acquisition, Methodology, Project administration, Writing – review and editing; Guixiu Shi, Funding acquisition, Project administration, Writing – review and editing

### Author ORCIDs

Yufei Jiang ⬤ https://orcid.org/0000-0002-0048-1241
Haibin Wang ⬤ http://orcid.org/0000-0002-9865-324X
Shuangbo Kong ⬤ https://orcid.org/0000-0002-7513-4041

### Decision letter and Author response

Decision letter https://doi.org/10.7554/eLife.83083.sa1
Author response https://doi.org/10.7554/eLife.83083.sa2

## Additional files

### Supplementary files

• MDAR checklist

• Supplementary file 1. Tables of characteristics of all patients and controls, all primers, inhibitors and activators used in this study.

### Data availability

Sequence data have been deposited in the National Center for Biotechnology Information database under accession ID: PRJNA791538. All other data associated with this study are present in the paper or the supplementary materials.

The following dataset was generated:

| Author(s) | Year | Dataset title | Dataset URL | Database and Identifier |
|-----------|------|---------------|-------------|------------------------|
| Jiang Y | 2022 | Global studies of regulation during decidualization | https://www.ncbi.nlm.nih.gov/bioproject/?term=PRJNA791538 | NCBI BioProject, PRJNA791538 |

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
