## [Editor Report]

This study presents a valuable finding on the role and function of the Gaq axis on the inflammatory response during decidualization essential for early pregnancy. The evidence supporting the claims of the authors is solid and clarified. The work will be of interest to reproductive biologists and clinicians.

---

## [Decision Letter]

**Decision letter after peer review:**

Thank you for submitting your article "Gaq-PKD/PKCμ regulates the IkB transcription to limit the NF-κB mediated inflammatory response essential for early pregnancy" for consideration by *eLife*. Your article has been reviewed by 2 peer reviewers, and the evaluation has been overseen by a Reviewing Editor and Ricardo Azziz as the Senior Editor. The reviewers have opted to remain anonymous.

Essential revisions:

1) Although all the experiments were well-designed and performed, the description is not clear enough to explain what they found. First of all, professional English editing is strongly recommended to provide a sound explanation for all the data in the manuscript. Many lines contain inappropriate words, typing errors, and grammar errors in the submitted manuscript. For example, in line 38 (in the abstract), HADC5 should be changed to HDAC5. In line 126, mid-proliferate should be changed to mid-proliferative. In line 273, "G protein Gs" what does it mean?

2) Whereas the data of Western blotting analyses in Figures 2 and 3 are convincing, it is not easy to see what they found in the immunofluorescence staining data in these figures. The authors are recommended to change the images for better understanding.

3) There are no statistical analyses for the role of stromal Gaq for blastocyst hatching and adhesion in a co-culture assay in Figure 6. Furthermore, considering the blastocyst number used in this experiment, the experiments seem to be performed once or twice. The authors are recommended to provide statistics for this experiment and to increase the number of the experiment. It is better to have data with statistical significance as graphs in this figure as well.

4) The authors need to provide a scientific rationale on why the authors stick to only HDAC5 as downstream of PKD/PKCu. They simply mentioned that "PKD/PKCu could directly phosphorylate HDAC5" before they describe the results in the result section.

Furthermore, it is not clear why the authors did not include the word, HDAC5, in the tile. Since HDAC5 is a key molecule of this manuscript, HDAC5 is supposed to be included in the title.

5) Lines 170-173: The investigators note that their findings using the KEGG analysis "were highly related to decidualization". However, Figure 1F does not denote 'decidualization' as a category. Please restate and explain the statement.

6) Lines 183-185: The investigators note that "As shown in Figure 2A-C and Figure supplement 3A, nuclear translocation and phosphorylation of the NF-κB subunit p65 and p50 were higher in GNAQ-KO decidual cells than in control." However, examination of the figure does not indicate this finding. Please address.

7) Lines 195-198: The authors state that "We performed western blot to analyze the STAT3 phosphorylation level in decidual cells upon Gaq knockout, and found that Gaq deficiency could significantly enhance the phosphorylation of STAT3 in decidual stromal cells (Figure 2F)". What were the results?

8) Lines 219-222: The authors state that "Interestingly, we found that both mRNA and protein level of NF.BIA was significantly attenuated in GNAQ-KO decidual cells, but not in Gaq deficiency proliferative stromal compared with control cells (Figure 3A, B)." On examination of the figure, this reviewer could not see any difference in the blots, except maybe in D2 and D4. Can the authors address this?

9) Lines 226-228: The authors state "As expected, nuclear translocation of the NF-κB subunit p65 and p50 were less in NFκBIA overexpression GNAQ-KO decidual cells (Figure 3C)". Again, this reviewer does not see this difference in the figure cited.

10) Line 263: The authors note the response of the PKD/PKCμ pathway to 1 nM PMA. Was there any attempt to determine whether there was a dose-dependent response.

11) Lines 502-504: The authors state that "The purity of stromal cells was verified by immunofluorescence staining with antibodies against the stromal-specific vimentin at the third passage." Which antibodies?

12) Lines 664-675: Regarding the study of blastocysts, this reviewer could not find any details regarding the source of these in the Materials and methods. Human or murine? This needs to be clearly discussed in the Materials and methods and in the results.

13) In the figures the authors repeatedly state that "Representative data are shown from three to five independent experiments." Could the authors clarify exactly what this means? Wasn't all data analyzed? Also, were the experiments performed in duplicate or triplicate? This needs to be noted.

14) In the manuscript (page 51 of merged PDF), part A (top) of Figure Supplement 3 appears to be a duplicate of Figure 1F and is different from the same figure at the end of the PDF (page 64).

*Reviewer #1 (Recommendations for the authors):*

1) Lines 170-173: The investigators note that their findings using the KEGG analysis "were highly related to decidualization". However, Figure 1F does not denote 'decidualization' as a category. Please restate and explain the statement.

2) Lines 183-185: The investigators note that "As shown in Figure 2A-C and Figure supplement 3A, nuclear translocation and phosphorylation of the NF-κB subunit p65 and p50 were higher in GNAQ-KO decidual cells than in control." However, examination of the figure does not indicate this finding. Please address.

3) Lines 195-198: The authors state that "We performed western blot to analyze the STAT3 phosphorylation level in decidual cells upon Gaq knockout, and found that Gaq deficiency could significantly enhance the phosphorylation of STAT3 in decidual stromal cells (Figure 2F)". What were the results?

4) Lines 219-222: The authors state that "Interestingly, we found that both mRNA and protein level of NF.BIA was significantly attenuated in GNAQ-KO decidual cells, but not in Gaq deficiency proliferative stromal compared with control cells (Figure 3A, B)." On examination of the figure, this reviewer could not see any difference in the blots, except maybe in D2 and D4. Can the authors address this?

5) Lines 226-228: The authors state "As expected, nuclear translocation of the NF-κB subunit p65 and p50 were less in NFκBIA overexpression GNAQ-KO decidual cells (Figure 3C)". Again, this reviewer does not see this difference in the figure cited.

6) Line 263: The authors note the response of the PKD/PKCμ pathway to 1 nM PMA. Was there any attempt to determine whether there was a dose-dependent response?

7) Lines 502-504: The authors state that "The purity of stromal cells was verified by immunofluorescence staining with antibodies against the stromal-specific vimentin at the third passage." Which antibodies?

8) Lines 664-675: Regarding the study of blastocysts, this reviewer could not find any details regarding the source of these in the Materials and methods. Human or murine? This needs to be clearly discussed in the Materials and methods and in the results.

9) In the figures the authors repeatedly state that "Representative data are shown from three to five independent experiments." Could the authors clarify exactly what this means? Wasn't all data analyzed? Also, were the experiments performed in duplicate or triplicate? This needs to be noted.

10) In the manuscript (page 51 of merged PDF), part A (top) of Figure Supplement 3 appears to be a duplicate of Figure 1F and is different from the same figure at the end of the PDF (page 64).

*Reviewer #2 (Recommendations for the authors):*

Jiang et al. performed a series of experiments to answer fundamental questions on the function of Gaq during decidualization of human endometrial stromal cells in vitro and in vivo. Experiments were well-designed and performed to answer sequential scientific questions using various molecular and

pharmacologic tools. However, several major concerns must be met before the publication in *eLife*.

1. Although all the experiments were well-designed and performed, the description is not clear enough to explain what they found. First of all, professional English editing is strongly recommended to provide a sound explanation for all the data in the manuscript. Many lines contain inappropriate words, typing errors, and grammar errors in the submitted manuscript. For example, in line 38 (in the abstract), HADC5 should be changed to HDAC5. In line 126, mid-proliferate should be changed to mid-proliferative. In line 273, "G protein Gs" what does it mean?

2. Whereas the data of Western blotting analyses in Figures 2 and 3 are convincing, it is not easy to see what they found in the immunofluorescence staining data in these figures. The authors are recommended to change the images for better understanding.

3. There are no statistical analyses for the role of stromal Gaq for blastocyst hatching and adhesion in a co-culture assay in Figure 6. Furthermore, considering the blastocyst number used in this experiment, the experiments seem to be performed once or twice. The authors are recommended to provide statistics for this experiment and to increase the number of the experiment. It is better to have data with statistical significance as graphs in this figure as well.

4. The authors need to provide a scientific rationale on why the authors stick to only HDAC5 as downstream of PKD/PKCu. They simply mentioned that "PKD/PKCu could directly phosphorylate HDAC5" before they describe the results in the result section.

Furthermore, it is not clear why the authors did not include the word, HDAC5, in the tile. Since HDAC5 is a key molecule of this manuscript, HDAC5 is supposed to be included in the title.

[Editors' note: further revisions were suggested prior to acceptance, as described below.]

Thank you for resubmitting your work entitled "Gαq-PKD/PKCμ signal regulating the nuclear export of HDAC5 to induce the IκB expression and limit the NF-κB mediated inflammatory response essential for early pregnancy" for further consideration by *eLife*. Your revised article has been evaluated by Diane Harper (Senior Editor) and a Reviewing Editor.

The manuscript has been improved but there are some remaining issues that need to be addressed, as outlined below by Reviewer #3:

1. In this work, the Authors have designed and performed the experiments in a well-defined manner. But in some cases the explanations need to be more clear and there are many typographical errors which need to be corrected. For example, using of different denotations like Gaq and Gαq of the same gene or protein (use single notation throughout the manuscript like either Gaq or Gαq).

2. In Line 154: The authors have mentioned the generation of GNAQ-KO cell lines through CRISPR. But a detailed explanation is required like the size of GNAQ gene (I will suggest doing a Schematic representation for the generation of the cell line).

3. Authors also mentioned that inserted mutation in both GNAQ alleles (line 155). If the cell lines are generated through inserting a mutation, what kind of a mutation and the location of the mutation and justification for the mutation site needs to be addressed? Knocking out a gene and inserting a mutation are two different strategies.

4. The authors have well represented the western blots with the loading controls. But in some figures like Figure 1E and Figure 2C loading controls of GAPDH and H3 are not comparable (not matching with) with the original blot. Original blot images without adjusting the contrast so much is needed for clarity.

5. In Figure 3, the Authors have mentioned an empty vector and in the legend it's mentioned as a control lentivirus. Are these both similar? If yes, it needs to be corrected as lentivirus transfected with empty vector.

6. Source of lentiviral vector (catalogue no.) needs to be mentioned.

7. In Line 247, the Authors have mentioned reintroducing the HA Tag. What does it mean by (like authors already have existing HA tagged Gαq, they removed it for some studies and they are reintroducing it again) or adding HA tag to WT- Gαq, details are required. Explain the details of how the amplification/cloning is performed.

8. HA-tagged WT Gαq can be denoted as HA-Gαq instead OF GNAQ-WT IN THE FIGURE LEGENDS (Figure 4).

9. Authors have performed the ChIP for localization studies (Figure 4). The amount of protein loaded looks quite high. Please mention the amount loaded and also the total amount of protein from Chip Assay along with the source material (starting material). The loading controls needs to be verified in Figure 4F.

10. Authors mentioned cytoplastic localization of Gαq. What does it mean by like Gαq is a cytoplasmic protein.

11. According to the western Blot results (Figure 4a)- Gαq is more widely expressed in nucleoplasm than cytoplasm and it is a cytoplasmic protein. A detailed explanation is required why it is so.

12. In Figure 5, the Expression of PKCα/β data is not shown. As the Authors are describing the expression of genes so the quantification of western blots is required.

13. References are required for the methodologies mentioned like Nuclear Protein Extracts and Chromatin protein precipitation.

---

## [Author Response]

Essential revisions:1) Although all the experiments were well-designed and performed, the description is not clear enough to explain what they found. First of all, professional English editing is strongly recommended to provide a sound explanation for all the data in the manuscript. Many lines contain inappropriate words, typing errors, and grammar errors in the submitted manuscript. For example, in line 38 (in the abstract), HADC5 should be changed to HDAC5. In line 126, mid-proliferate should be changed to mid-proliferative. In line 273, "G protein Gs" what does it mean?

We have checked the language and grammar of our manuscript by a professional expert and all authors.

Thanks for the reviewer’s concern to strength our manuscript. We have changed HDAC5 to HADC5, and mid-proliferative to mid-proliferate in revised manuscript.

Sorry for the unclear description we used in this study, there are three major G protein subfamilies that regulate distinct cellular signaling pathways: adenylyl cyclase stimulatory G protein (Gs), Gi/o, or Gq/11. We have rephrased these sentences as “G protein adenylyl cyclase stimulatory G protein (Gs)” in revised manuscript.

2) Whereas the data of Western blotting analyses in Figures 2 and 3 are convincing, it is not easy to see what they found in the immunofluorescence staining data in these figures. The authors are recommended to change the images for better understanding.

As suggested, we changed the immunofluorescence staining data for Figures 2 and 3 in revised manuscript, and we also provided the data of Western blotting quantification analyses (Revised Figure2—figure supplement 2 and Figure3—figure supplement 1) and description.

3) There are no statistical analyses for the role of stromal Gaq for blastocyst hatching and adhesion in a co-culture assay in Figure 6. Furthermore, considering the blastocyst number used in this experiment, the experiments seem to be performed once or twice. The authors are recommended to provide statistics for this experiment and to increase the number of the experiment. It is better to have data with statistical significance as graphs in this figure as well.

Many thanks for this kind suggestion. It’s hard for us to manipulate many blastocysts for one experiment, so, we performed five experiments to get the data for enough blastocysts in this experiment. We have added number of the experiments, statistical analyses and description in revised figure 6C for the role of stromal Gaq for blastocyst hatching and adhesion in the co-culture assay.

4) The authors need to provide a scientific rationale on why the authors stick to only HDAC5 as downstream of PKD/PKCu. They simply mentioned that "PKD/PKCu could directly phosphorylate HDAC5" before they describe the results in the result section.Furthermore, it is not clear why the authors did not include the word, HDAC5, in the tile. Since HDAC5 is a key molecule of this manuscript, HDAC5 is supposed to be included in the title.

Very kind suggestion. The title was changed into a new one which include HDAC5. We also added a description for the scientific rationale to detect the HDAC5 in the revised manuscript as follows: Gene transcription can be regulated by the transcriptional factor and epigenetic mechanism (PMID:31813015). As one of the epigenetic modifications, the histone acetylation can induce the target gene transcription. It has been reported that PKD/PKCμ could directly phosphorylated HDAC5, one of the deacetylases.

5) Lines 170-173: The investigators note that their findings using the KEGG analysis "were highly related to decidualization". However, Figure 1F does not denote 'decidualization' as a category. Please restate and explain the statement.

Thanks for your suggestion, we have restated and explained the statement "included the Wnt signaling pathway (PMID: 22115959, PMID: 29259425) and JAK-STAT, NFκB inflammation pathway, which had been reported to regulate stromal apoptosis and differentiation during the decidualization" in revised manuscript.

6) Lines 183-185: The investigators note that "As shown in Figure 2A-C and Figure supplement 3A, nuclear translocation and phosphorylation of the NF-κB subunit p65 and p50 were higher in GNAQ-KO decidual cells than in control." However, examination of the figure does not indicate this finding. Please address.

Thank for the reviewer’s good suggestion and we have changed the immunofluorescence staining data for Figure 2 and Figure 2—figure supplement 1A, and also provided the data of Western blotting quantification analyses to confirm the change at protein level (see revised Figure2—figure supplement 2).

7) Lines 195-198: The authors state that "We performed western blot to analyze the STAT3 phosphorylation level in decidual cells upon Gaq knockout, and found that Gaq deficiency could significantly enhance the phosphorylation of STAT3 in decidual stromal cells (Figure 2F)". What were the results?

As suggested, we rearranged the data of Western blotting analyses of p-STAT3 (Revised Figure2—figure supplement 2E) and description in revised manuscript.

8) Lines 219-222: The authors state that "Interestingly, we found that both mRNA and protein level of NF.BIA was significantly attenuated in GNAQ-KO decidual cells, but not in Gaq deficiency proliferative stromal compared with control cells (Figure 3A, B)." On examination of the figure, this reviewer could not see any difference in the blots, except maybe in D2 and D4. Can the authors address this?

Many thanks for this kind suggestion. As we stated in discussion “Gαq deficiency attenuated NFκBIA mRNA expression in early decidual cells (D2, D4), but not late decidual cells and proliferative stromal cells”, so, we have added the word” early” to the revised manuscript to improve its accurate description.

9) Lines 226-228: The authors state "As expected, nuclear translocation of the NF-κB subunit p65 and p50 were less in NFκBIA overexpression GNAQ-KO decidual cells (Figure 3C)". Again, this reviewer does not see this difference in the figure cited.

As suggested, we have changed the immunofluorescence staining data for Figures 2 and 3 in revised figures, added the description in the manuscript.

10) Line 263: The authors note the response of the PKD/PKCμ pathway to 1 nM PMA. Was there any attempt to determine whether there was a dose-dependent response.

We appreciate the reviewer’s good suggestion. PMA as a PKC agonist, at the concentration of 100nM (recommended concentration) was able to mimics the DAG to activate PKC. In the present work, PMA at the low concentration of 1nM can activate PKD/PKCμ in Gαq deficiency and normal decidual cells. There was no obvious dose-dependent response in phosphorylation of PKD/PKC at the concentrations of 2nM, 5nM and 20nM. But the phosphorylation of PKD/PKC was inhibited at 20nM. Furthermore, PMA could inhibit total protein level of PKD/PKC at the concentrations of 5nM and 20nM.

11) Lines 502-504: The authors state that "The purity of stromal cells was verified by immunofluorescence staining with antibodies against the stromal-specific vimentin at the third passage." Which antibodies?

Very sorry for the inaccurate description. Vimentin (Cat# ab92547, Abcam, USA) just the antibody against stromal-specific vimentin, which negative in epithelial cells. We have corrected the text in revised manuscript (See Page 20).

12) Lines 664-675: Regarding the study of blastocysts, this reviewer could not find any details regarding the source of these in the Materials and methods. Human or murine? This needs to be clearly discussed in the Materials and methods and in the results.

Thank for the reviewer’s good suggestion, all the blastocysts used in the work from human abandoned blastocysts and we have added clearly details in revised manuscript.

13) In the figures the authors repeatedly state that "Representative data are shown from three to five independent experiments." Could the authors clarify exactly what this means? Wasn't all data analyzed? Also, were the experiments performed in duplicate or triplicate? This needs to be noted.

Sorry for the unapparent description, we performed at least three or five independent experiments and all date were analyzed. We have written the information in each figure legend in revised manuscript.

14) In the manuscript (page 51 of merged PDF), part A (top) of Figure Supplement 3 appears to be a duplicate of Figure 1F and is different from the same figure at the end of the PDF (page 64).

Very sorry for our carelessness. We have uploaded the same figure in revised manuscript.

Reviewer #1 (Recommendations for the authors):1) Lines 170-173: The investigators note that their findings using the KEGG analysis "were highly related to decidualization". However, Figure 1F does not denote 'decidualization' as a category. Please restate and explain the statement.

Thanks for your suggestion, we have restated and explained the statement "included the Wnt signaling pathway (PMID: 22115959, PMID: 29259425) and JAK-STAT, NFκB inflammation pathway, which had been reported to regulate stromal apoptosis and differentiation during the decidualization" in revised manuscript.

2) Lines 183-185: The investigators note that "As shown in Figure 2A-C and Figure supplement 3A, nuclear translocation and phosphorylation of the NF-κB subunit p65 and p50 were higher in GNAQ-KO decidual cells than in control." However, examination of the figure does not indicate this finding. Please address.

Thank for the reviewer’s good suggestion and we have changed the immunofluorescence staining data for Figure 2 and Figure2—figure supplement 1A, and also provided the data of Western blotting quantification analyses to confirm the change at protein level (see revised Figure2—figure supplement 2).

3) Lines 195-198: The authors state that "We performed western blot to analyze the STAT3 phosphorylation level in decidual cells upon Gaq knockout, and found that Gaq deficiency could significantly enhance the phosphorylation of STAT3 in decidual stromal cells (Figure 2F)". What were the results?

As suggested, we rearranged the data of Western blotting analyses of p-STAT3 (Revised Figure2—figure supplement 2E) and description in revised manuscript.

4) Lines 219-222: The authors state that "Interestingly, we found that both mRNA and protein level of NF.BIA was significantly attenuated in GNAQ-KO decidual cells, but not in Gaq deficiency proliferative stromal compared with control cells (Figure 3A, B)." On examination of the figure, this reviewer could not see any difference in the blots, except maybe in D2 and D4. Can the authors address this?

Many thanks for this kind suggestion. As we stated in discussion “Gαq deficiency attenuated NFκBIA mRNA expression in early decidual cells (D2, D4), but not late decidual cells and proliferative stromal cells”, so, we have added the word” early” to the revised manuscript to improve its accurate description.

5) Lines 226-228: The authors state "As expected, nuclear translocation of the NF-κB subunit p65 and p50 were less in NFκBIA overexpression GNAQ-KO decidual cells (Figure 3C)". Again, this reviewer does not see this difference in the figure cited.

As suggested, we have changed the immunofluorescence staining data for Figures 2 and 3 in revised figures, added the description in the manuscript.

6) Line 263: The authors note the response of the PKD/PKCμ pathway to 1 nM PMA. Was there any attempt to determine whether there was a dose-dependent response?

We appreciate the reviewer’s good suggestion. PMA as a PKC agonist, at the concentration of 100 nM (recommended concentration) was able to mimics the DAG to activate PKC. In the present work, PMA at the low concentration of 1nM could activate PKD/PKCμ in Gαq deficiency and normal decidual cells. There was no obvious dose-dependent response in phosphorylation of PKD/PKC at the concentrations of 2nM, 5nM and 20nM. But the phosphorylation of PKD/PKC was inhibited at 20nM. Furthermore, PMA could inhibit total protein level of PKD/PKC at the concentrations of 5nM and 20nM.

7) Lines 502-504: The authors state that "The purity of stromal cells was verified by immunofluorescence staining with antibodies against the stromal-specific vimentin at the third passage." Which antibodies?

Very sorry for the inaccurate description. Vimentin (Cat# ab92547, Abcam, USA) just the antibody against stromal-specific vimentin, which negative in epithelial cells. We have corrected the text in revised manuscript (See Page 20).

8) Lines 664-675: Regarding the study of blastocysts, this reviewer could not find any details regarding the source of these in the Materials and methods. Human or murine? This needs to be clearly discussed in the Materials and methods and in the results.

Thank for the reviewer’s good suggestion, all the blastocysts used in the work from human abandoned blastocysts and we have added clearly details in revised manuscript.

9) In the figures the authors repeatedly state that "Representative data are shown from three to five independent experiments." Could the authors clarify exactly what this means? Wasn't all data analyzed? Also, were the experiments performed in duplicate or triplicate? This needs to be noted.

Sorry for the unapparent description, we performed at least three or five independent experiments and all date were analyzed. We have written the information in each figure legend in revised manuscript.

10) In the manuscript (page 51 of merged PDF), part A (top) of Figure Supplement 3 appears to be a duplicate of Figure 1F and is different from the same figure at the end of the PDF (page 64).

Very sorry for our carelessness. We have uploaded the same figure in revised manuscript.

Reviewer #2 (Recommendations for the authors):Jiang et al. performed a series of experiments to answer fundamental questions on the function of Gaq during decidualization of human endometrial stromal cells in vitro and in vivo. Experiments were well-designed and performed to answer sequential scientific questions using various molecular and pharmacologic tools. However, several major concerns must be met before the publication in eLife.1. Although all the experiments were well-designed and performed, the description is not clear enough to explain what they found. First of all, professional English editing is strongly recommended to provide a sound explanation for all the data in the manuscript. Many lines contain inappropriate words, typing errors, and grammar errors in the submitted manuscript. For example, in line 38 (in the abstract), HADC5 should be changed to HDAC5. In line 126, mid-proliferate should be changed to mid-proliferative. In line 273, "G protein Gs" what does it mean?

We have checked the language and grammar of our manuscript by a professional expert and all authors.

Thanks for the reviewer’s concern to strength our manuscript. We have changed HDAC5 to HADC5, and mid-proliferative to mid-proliferate in revised manuscript.

Sorry for the unclear description we used in this study, there are three major G protein subfamilies that regulate distinct cellular signaling pathways: adenylyl cyclase stimulatory G protein (Gs), Gi/o, or Gq/11. We have rephrased these sentences as “G protein adenylyl cyclase stimulatory G protein (Gs)” in revised manuscript.

2. Whereas the data of Western blotting analyses in Figures 2 and 3 are convincing, it is not easy to see what they found in the immunofluorescence staining data in these figures. The authors are recommended to change the images for better understanding.

As suggested, we changed the immunofluorescence staining data for Figures 2 and 3 in revised manuscript, and we also provided the data of Western blotting quantification analyses (Revised Figure2—figure supplement 2 and Figure3—figure supplement 1) and description.

3. There are no statistical analyses for the role of stromal Gaq for blastocyst hatching and adhesion in a co-culture assay in Figure 6. Furthermore, considering the blastocyst number used in this experiment, the experiments seem to be performed once or twice. The authors are recommended to provide statistics for this experiment and to increase the number of the experiment. It is better to have data with statistical significance as graphs in this figure as well.

Many thanks for this kind suggestion. It’s hard for us to manipulate many blastocysts for one experiment, so, we performed five experiments to get the data for enough blastocysts in this experiment. We have added number of the experiments, statistical analyses and description in revised figure 6C for the role of stromal Gaq for blastocyst hatching and adhesion in the co-culture assay.

4. The authors need to provide a scientific rationale on why the authors stick to only HDAC5 as downstream of PKD/PKCu. They simply mentioned that "PKD/PKCu could directly phosphorylate HDAC5" before they describe the results in the result section.Furthermore, it is not clear why the authors did not include the word, HDAC5, in the tile. Since HDAC5 is a key molecule of this manuscript, HDAC5 is supposed to be included in the title.

Very kind suggestion. The title was changed into a new one which include HDAC5. We also added a description for the scientific rationale to detect the HDAC5 in the revised manuscript as follows: Gene transcription can be regulated by the transcriptional factor and epigenetic mechanism (PMID:31813015). As one of the epigenetic modifications, the histone acetylation can induce the target gene transcription. It has been reported that PKD/PKCμ could directly phosphorylated HDAC5, one of the deacetylases.

[Editors' note: further revisions were suggested prior to acceptance, as described below.]

The manuscript has been improved but there are some remaining issues that need to be addressed, as outlined below by Reviewer #3:1. In this work, the Authors have designed and performed the experiments in a well-defined manner. But in some cases the explanations need to be more clear and there are many typographical errors which need to be corrected. For example, using of different denotations like Gaq and Gαq of the same gene or protein (use single notation throughout the manuscript like either Gaq or Gαq).

Many thanks for this kind suggestion. Sorry for our carelessness. We have changed all typographical errors in the revised manuscript.

2. In Line 154: The authors have mentioned the generation of GNAQ-KO cell lines through CRISPR. But a detailed explanation is required like the size of GNAQ gene (I will suggest doing a Schematic representation for the generation of the cell line).

Very kind suggestion. We have drawn a new schematic showing the generation of the GNAQ-KO cell line show in the revised Figure 1—figure supplement 2.

3. Authors also mentioned that inserted mutation in both GNAQ alleles (line 155). If the cell lines are generated through inserting a mutation, what kind of a mutation and the location of the mutation and justification for the mutation site needs to be addressed? Knocking out a gene and inserting a mutation are two different strategies.

We appreciate the reviewer’s concern. In present study, Cas9-generated double strand DNA breaks were repaired through Non-Homologous End Joining (NHEJ), which is prone to generate insertion or deletion mutations, and thus cause the translational frameshift to destroy the production of functional protein. The homologue repaired plasmid was not utilized in this system. In this study, the CRISPR-Cas9 system result in Gαq knockout harboring a single base insertion near the Cas9 cutting site, which lead to a frameshift in the protein coding region (revised Figure 1—figure supplement 2). We have restated and explained the statement "Sanger sequencing proved that the clonal cell line harbors a single base insertion near the Cas9 cutting site, causing a frameshift in the protein coding region" in the revised manuscript.

4. The authors have well represented the western blots with the loading controls. But in some figures like Figure 1E and Figure 2C loading controls of GAPDH and H3 are not comparable (not matching with) with the original blot. Original blot images without adjusting the contrast so much is needed for clarity.

We are grateful for the reviewer's concern and feedback. We re-checked data and confirmed that the original blot images of Figure 1—figure supplement 1 loading controls of GAPDH and H3 in Figure 1—figure supplement 1-source data, along with their corresponding markings "Figure1—figure supplement 1E GAPDH" and "Figure1—figure supplement 1E H3", as well as the original blot image of Figure 2C loading controls of GAPDH in Figure 2-source data marked with "Figure 2C GAPDH", were provided without any contrast adjustments.

5. In Figure 3, the Authors have mentioned an empty vector and in the legend it's mentioned as a control lentivirus. Are these both similar? If yes, it needs to be corrected as lentivirus transfected with empty vector.

Sorry for the unapparent description, “empty vector” or “control lentivirus” used in the Figure 3 are both similar, as suggested, we have changed “empty vector” or “control lentivirus” to “lentivirus transfected with empty vector” in the revised manuscript.

6. Source of lentiviral vector (catalogue no.) needs to be mentioned.

Thank for the reviewer’s good suggestion and we have added this important information (Takara, catalogue no.: 632187) for lentiviral vector used in our study in Materials and methods in the revised manuscript.

7. In Line 247, the Authors have mentioned reintroducing the HA Tag. What does it mean by (like authors already have existing HA tagged Gαq, they removed it for some studies and they are reintroducing it again) or adding HA tag to WT- Gαq, details are required. Explain the details of how the amplification/cloning is performed.

Sorry for the unclear description we used in this study. In this study, we aimed to evaluate the function of Gαq in different compartments (nucleoplasm and cytoplasm) by introducing HA-Gαq or GNAQ-NES plasmids that contained a potent nuclear export sequence into Gαq knockout decidual cells. We added an HA tag to WT-Gαq by adding the HA sequence (AGCGTAATCTGGAACATCGTATGGGTA) to the C terminal of the GNAQ coding sequence as a reverse primer (AGCGTAATCTGGAACATCGTATGGGTA(HA)GACCAGATTGTACTCCTTCAGGTTCAAC). We amplified the GNAQ coding sequence by PCR and cloned it into the pLVX-IRES-ZsGreen lentiviral plasmids, which were then co-transfected into HEK293T cells with lentivirus genomic plasmids for lentiviral packaging. We confirmed all clones by DNA sequencing and analyzed them for mRNA and protein expression to confirm overexpression efficiency.

8. HA-tagged WT Gαq can be denoted as HA-Gαq instead OF GNAQ-WT in the figure legends (Figure 4).

Very kind suggestion. We would like to note that in the revised version of Figure 4, we have denoted HA-Gαq instead of GNAQ-WT in the figure legends.

9. Authors have performed the ChIP for localization studies (Figure 4). The amount of protein loaded looks quite high. Please mention the amount loaded and also the total amount of protein from Chip Assay along with the source material (starting material). The loading controls needs to be verified in Figure 4F.

Thank for the reviewer’s concern, a quantity of 15μg cytoplasm was loaded, while 50μl nucleoplasm or 50μl chromatin were loaded without being measured. These samples were used as loading controls in Figure 4 to demonstrate the distribution of Gαq in various subcellular fractions of HESC (See revised manuscript line 656).

10. Authors mentioned cytoplastic localization of Gαq. What does it mean by like Gαq is a cytoplasmic protein.

Sorry for any confusion caused by unclear description. “cytoplastic localization of Gαq” it means Gαq displayed the cytoplasmic localization in HESC, evidence from immunofluorescence analyses and subcellular fraction experiments were performed.

11. According to the western Blot results (Figure 4a)- Gαq is more widely expressed in nucleoplasm than cytoplasm and it is a cytoplasmic protein. A detailed explanation is required why it is so.

Based on our data of both immunofluorescence and subcellular fraction analyses, Gαq can localize both in nucleoplasm and cytoplasm, and we did not define Gαq as a cytoplasmic protein. In consistent with our data, its nucleoplasm localization has been reported in other published papers (PMID: 25590750, PMID: 16221676).

12. In Figure 5, the Expression of PKCα/β data is not shown. As the Authors are describing the expression of genes so the quantification of western blots is required.

As suggested, we have added data of Western blotting quantification of p-PKCα/β and PKCα (See Revised Figure 5—figure supplement 2 and revised manuscript Line 1155).

13. References are required for the methodologies mentioned like Nuclear Protein Extracts and Chromatin protein precipitation.

Very kind suggestion. We have added references for the methodologies mentioned Nuclear Protein Extracts and Chromatin protein precipitation in revised manuscript (See revised manuscript lines 643 and 657).